# Elucidating the catalytic mechanism of Prussian blue nanozymes with self-increasing catalytic activity

Kaizheng Feng[1], Zhenzhen Wang[2], Shi Wang[1], Guancheng Wang[1], Haijiao Dong [3], Hongliang He[1], Haoan Wu[1], Ming Ma [1] ✉, Xingfa Gao [2] ✉ & Yu Zhang [1] ✉

Although Prussian blue nanozymes (PBNZ) are widely applied in various fields, their catalytic mechanisms remain elusive. Here, we investigate the long-term catalytic performance of PBNZ as peroxidase (POD) and catalase (CAT) mimetics to elucidate their lifespan and underlying mechanisms. Unlike our previously reported $Fe_3O_4$ nanozymes, which exhibit depletable POD-like activity, the POD and CAT-like activities of PBNZ not only persist but slightly enhance over prolonged catalysis. We demonstrate that the irreversible oxidation of PBNZ significantly promotes catalysis, leading to self-increasing catalytic activities. The catalytic process of the pre-oxidized PBNZ can be initiated through either the conduction band pathway or the valence band pathway. In summary, we reveal that PBNZ follows a dual-path electron transfer mechanism during the POD and CAT-like catalysis, offering the advantage of a long service life.

Since the pioneering discovery of $Fe_3O_4$ nanoparticles as peroxidase (POD) memetics in 2007[1], nanozymes have garnered significant attention over the past decade[2–9]. Among the various research areas within nanozyme studies, investigating catalytic mechanisms is essential for the rational design and the application of nanozymes. Currently, single-atom nanozymes are widely selected for mechanism studies due to their well-defined and tunable structures[10,11]. Nevertheless, elucidating the mechanisms of traditional nanozymes with complex components and structures remains both challenging and crucial[12–18].

Since our group first reported the peroxidase (POD) and catalase (CAT)-like activities of Prussian blue nanozymes (PBNZ), these nanozymes have found wide application across various fields[19–23]. However, consensus on their catalytic mechanism remains elusive. For instance, our initial work speculated that PBNZ function as POD and CAT mimetics through a mechanism involving the conversion among Prussian blue (PB, $KFe(III)[Fe(II)(CN)_6]$), Berlin green (BG,

$KFe(III)_3[Fe(III)(CN)_6]_2[Fe(II)(CN)_6]$) and Prussian yellow (PY, $Fe(III)[Fe(III)(CN)_6]$), driven by matching redox potentials[24]. Dong's group highlighted the importance of N-coordinated Fe units of PBNZ for POD-like activity and proposed a catalytic mechanism similar to that of natural peroxidases, involving high-valent Fe (Fe (IV)=O) intermediate[25]. In contrast, Arkady's group suggested that PBNZ first react with a reductive substrate to form Prussian white (PW) and argued that the Fe (IV)=O-mediated mechanism is not applicable to PBNZ[26]. Additionally, several studies have attributed the POD-like activity of PBNZ to the generation of hydroxyl radical (·OH) and singlet oxygen ($^1O_2$)[27–32]. Despite this complexity, the overall electron flow direction of POD-like catalysis is from a reductive substrate (e.g., 2,2'-Azinobis (3-ethylbenzothiazoline −6-sulfonic Acid Ammonium Salt) (ABTS)) to $H_2O_2$. Two distinct electron transfer pathways could simultaneously exist: as a semiconductor[33], PB could undergo a valence band mediated pathway (VBP) where PB initially donates an electron to $H_2O_2$ (process 1) and then accepts another electron from

[1]State Key Laboratory of Digital Medical Engineering, Jiangsu Key Laboratory for Biomaterials and Devices, School of Biological Science and Medical Engineering & Basic Research and Innovation Center of Ministry of Education, Zhongda Hospital, Southeast University, Nanjing, China. [2]Laboratory of Theoretical and Computational Nanoscience, National Center for Nanoscience and Technology of China, Beijing, China. [3]Nanjing Institute of Measurement and Testing Technology, Nanjing, China. ✉e-mail: maming@seu.edu.cn; gaoxf@nanoctr.cn; zhangyu@seu.edu.cn

ABTS (process 2) (Fig. 1a). Alternatively, the catalytic reaction could proceed through a conduction band mediated pathway (CBP), where PB or their pre-oxidized state (BG, PY) first receive an electron from ABTS (process 1), followed by electron transfer to $H_2O_2$ (process 2) (Fig. 1b). From this perspective, current explanations for the catalytic mechanism of PBNZ can be attributed to these two catalytic pathways and a more explicit understanding of the mechanism is urgently needed.

Our previous work demonstrated the depletable POD-like activity of $Fe_3O_4$ nanozymes through long-term catalysis experiments[34]. Long-term catalysis is advantageous for accessing the life span of nanozymes and amplifying minor physicochemical changes that may be undetectable during the short-term catalysis. We believe this strategy is also beneficial for examining the catalytic sustainability of PBNZ, detecting subtle variations during the catalytic process, and elucidating their potential mechanisms.

In this work, we report the self-increasing catalytic activity of PBNZ observed during prolonged catalysis. The previously overlooked pre-oxidation of PBNZ during catalysis is responsible for the simultaneous facilitation of both the VBP and CBP, resulting in a dual-path electron transfer mechanism mediating the catalytic process of PBNZ.

## Results and discussion
### Characterization of PBNZ prepared by double injection method
PBNZ used for long-term catalysis in this work was prepared using our previously reported double injection method. By adjusting the volume of the $Fe^{3+}$-citric acid solution added initially (the $x$ value) and the reaction time, we could easily control the size and crystallinity of the PBNZ[35]. Dynamic light scattering (DLS), transmission electron microscopy (TEM), and ultraviolet-visible (UV-vis) spectroscopy confirmed the successful synthesis of PBNZ with sizes of 81 nm and 43 nm, exhibiting low crystallinity and good dispersity (Supplementary Figs. 1 and 2).

We then focused on the surficial composition of the as-prepared PBNZ. The X-ray photoelectron spectroscopy (XPS) spectra of the 81 nm and 43 nm PBNZ are shown in Fig. 2a, b, and Fig. 2d, e respectively. The $Fe2p_{3/2}$ peak of PBNZ can generally be deconvoluted into four peaks, similar to our previous work[36], which are assigned to C-coordinated Fe (II) at 708.6 eV, N-coordinated Fe (II) at 710.0 eV, N-coordinated Fe (III) at 712.8 eV and $Fe^{2+}$ satellite peak at 715.1 eV. Given that the PBNZ were prepared with an equivalent molar dosage of $Fe^{3+}$ and $[Fe(II)(CN)_6]^{4+}$, the presence of N-coordinated Fe (II) and low amount of Fe (III) on the surface of the PBNZ needs further clarification. Firstly, a similar $Fe2p$ spectrum was discovered in the PBNZ prepared by $Fe^{2+}$ and $[Fe(III)(CN)_6]^{3-}$ (Supplementary Fig. 3d). This suggests that the analogous $Fe2p$ spectra originate from the rapid valence state redistribution through inter-metal charge transfer among C-coordinated Fe and N-coordinated Fe once the PBNZ are formed. Additionally, the Fe:N ratio of 81 nm and 43 nm PBNZ detected by EDS and XPS analyses were lower than their theoretical formula $(KFe(III)[Fe(II)(CN)_6]$ (Fe:N = 1:3) or $Fe(III)_4[Fe(II)(CN)_6]_3$ (Fe:N = 1:2.6) (Fig. 2a, c, d, f)). This indicates a deficiency of $Fe^{3+}$ in the bulk phase and

surface of the as-prepared PBNZ due to the incomplete reaction of $Fe^{3+}$ with $[Fe(CN)_6]^{4+}$ during the ultra-fast synthesis. The insufficient use of $Fe^{3+}$ was further demonstrated by analyzing the components of the filtrate solution of the unpurified PBNZ (Supplementary Fig. 4). In summary, the relatively higher usage of $[Fe(CN)_6]^{4+}$ during the actual synthetic process led to the observed surficial coverage of excess C-coordinated Fe.

The specific catalytic activity ($a_{POD}$ for POD-like activity, $a_{CAT}$ for CAT-like activity) of 81 nm and 43 nm PBNZ was calculated. As expected, the 43 nm PBNZ outperformed the 81 nm PBNZ in the catalytic oxidation of ABTS (Fig. 2g), 3,3′,5,5′-tetramethylbenzidine (TMB) (Fig. 2h), and the self-decomposition of $H_2O_2$ (Fig. 2i). This is reflected in the higher $a_{POD}$ and $a_{CAT}$ values, demonstrating superior POD and CAT-like activity due to their larger specific surface area. Additionally, the high catalytic stability of the prepared PBNZ was confirmed (Supplementary Fig. 5).

### POD-like long-term catalysis of PBNZ
ABTS was chosen as the chromogenic substrate, and consecutive oxidation of ABTS catalyzed by PBNZ was observed over 120 h (Supplementary Fig. 6). Initially, ABTS undergoes a single electron transfer process to form green-colored $ABTS^+$ radicals, followed by a further electron transfer to generate colorless $ABTS^{2+}$[37]. This result indicates that the prepared PBNZ can maintain catalytic activity for up to 120 h, providing valuable insight for the parameter settings of subsequent cyclic catalysis experiments. As outlined in Fig. 3a, the cyclic catalysis was conducted in three rounds, with each round lasting 24 h. Remarkably, the POD-like activity of both 81 nm (Fig. 3b) and 43 nm PBNZ (Fig. 3c) increased with each successive round of catalysis. This continuous enhancement is notably different from the depletable POD-like activity observed in the cyclic catalysis of $Fe_3O_4$ nanozymes in previous study[34].

To investigate the underlying reason for the self-increasing POD-like activity of PBNZ, we carefully characterized the recycled PBNZ. Although the morphology of the collected PBNZ remained unchanged (Supplementary Fig. 7a, b), the hydrodynamic diameter of 81 nm and 43 nm PBNZ increased after three rounds of cyclic catalysis, indicating slight nanoparticle aggregation (Supplementary Fig. 7c, d). Additionally, we observed an increase in zeta potential and a slight decrease in absorbance in the adsorption spectra (Supplementary Fig. 7c–f). These findings suggest an irreversible oxidation process in which the surficial Fe (II) in PBNZ was oxidized to Fe (III) by $H_2O_2$ during the catalysis, leading to the decreased surface negative charge and reduced adsorption around 700 nm. XPS results further confirmed this speculation. During the cyclic catalysis, a certain amount of high-spin N-coordinated Fe (II) was oxidized by $H_2O_2$. Specifically, the surficial Fe (III) increased from 20.2% to 21.1% for 81 nm PBNZ and from 20.5% to 27.2% for 43 nm PBNZ, respectively (Fig. 3d, e). Although Dong's group reported undetected oxidation of PBNZ treated with 20 mM $H_2O_2$ for 60 min[25], our results revealed the actual existence of this irreversible oxidation process during PBNZ catalysis.

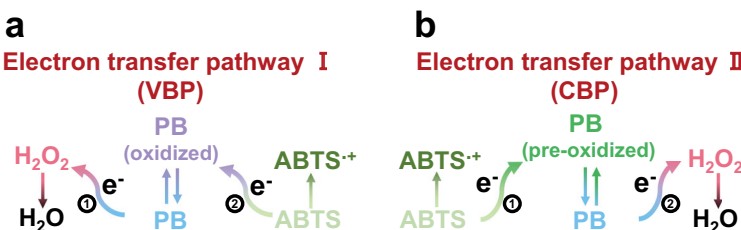

**a**
**Electron transfer pathway Ⅰ (VBP)**

**b**
**Electron transfer pathway Ⅱ (CBP)**

**Fig. 1 | Two possible electron transfer pathways during the POD-like catalysis of PBNZ. a** Electron transfer pathway I (VBP: valence band mediated pathway). **b** Electron transfer pathway II (CBP: conduction band mediated pathway).

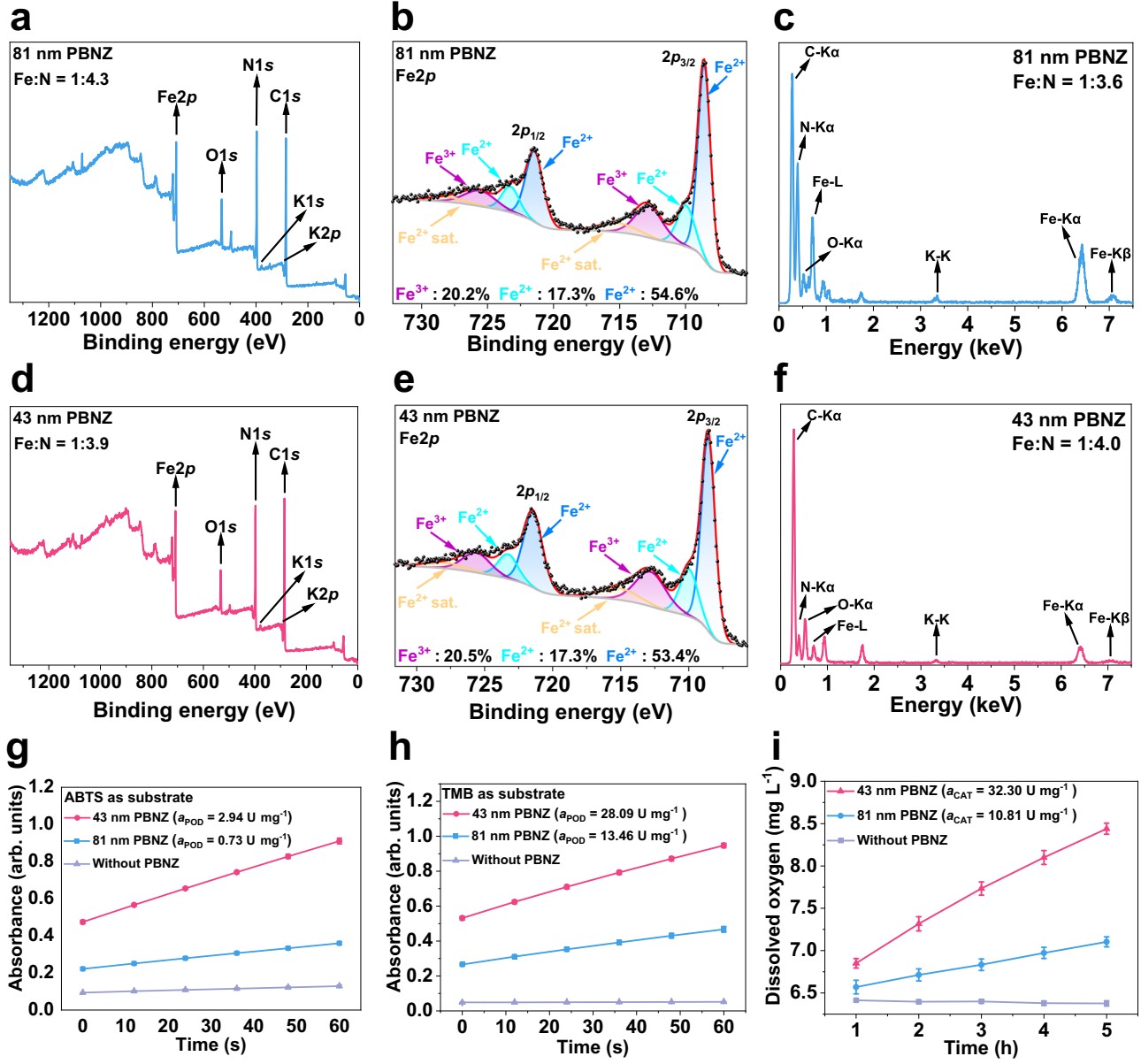

**Fig. 2 | Surface composition and catalytic activity of prepared PBNZ. a** XPS survey, **b** Fe2*p* spectrum, and **c** EDS spectrum of 81 nm PBNZ. **d** XPS survey, **e** Fe2*p* spectrum, and **f** EDS spectrum of 43 nm PBNZ. POD-like activity measurement using **g** ABTS and **h** TMB as substrates (n = 3 independent experiments). **i** CAT-like activity measurement (n = 3 independent experiments). Error bars represent standard deviation (SD) from three independent measurements and all data are presented as mean values ± SD.

Various control experiments for cyclic catalysis were conducted. When PBNZ were dispersed in HAc-NaAc buffer with or without 0.385 mg mL$^{-1}$ ABTS, slight aggregation and decreased catalytic activity were observed, while their morphology and zeta potential remained nearly unchanged (Supplementary Figs. 8 and 9). In contrast, when PBNZ were incubated with 0.12% $H_2O_2$ in HAc-NaAc buffer, there was a more significant decrease in surface negative charge and characteristic adsorption compared with the PBNZ recycled from cyclic catalysis (Supplementary Fig. 10). XPS results directly revealed further oxidation of PBNZ by $H_2O_2$, with the amount of surficial Fe (III) increasing to 24.3% and 29.2% for 81 nm (Fig. 3f) and 43 nm (Fig. 3g) PBNZ, respectively. As expected, the POD-like activity of the oxidized PBNZ showed a significant enhancement (Fig. 3h). This growing POD-like activity was also observed when using positively charged TMB as a chromogenic substrate, indicating that the increase in surficial Fe (III), rather than

the decrease in surface negative charge decrease, is the main reason for the enhanced catalytic activity (Fig. 3i).

To study the influence of crystallinity on the oxidation of PBNZ, highly crystalline PBNZ (denoted as 90 nm PBNZ) were synthesized with an *x* value of 1.0 mL and a reaction time of 2 h. These 90 nm PBNZ exhibited similar particle size, surficial valence state, and POD-like activity but differed in crystallinity compared with the 81 nm PBNZ (Supplementary Fig. 11a–f). Notably, the increment in Fe (III) and POD-like activity of 90 nm PBNZ after cyclic catalysis and incubation with 0.12% $H_2O_2$ were lower than that of the 81 nm PBNZ (Supplementary Fig. 11g–i). This suggests that low-crystallinity PBNZ are more easily oxidized due to their larger specific surface area and stronger interaction with $H_2O_2$, resulting in a higher increase in catalytic activity. Additionally, increased Fe leaching was observed during the cyclic catalysis and $H_2O_2$ incubation of PBNZ, indicating that Fe leaching could also reflect the oxidation of PBNZ (Supplementary Fig. 12). Taken

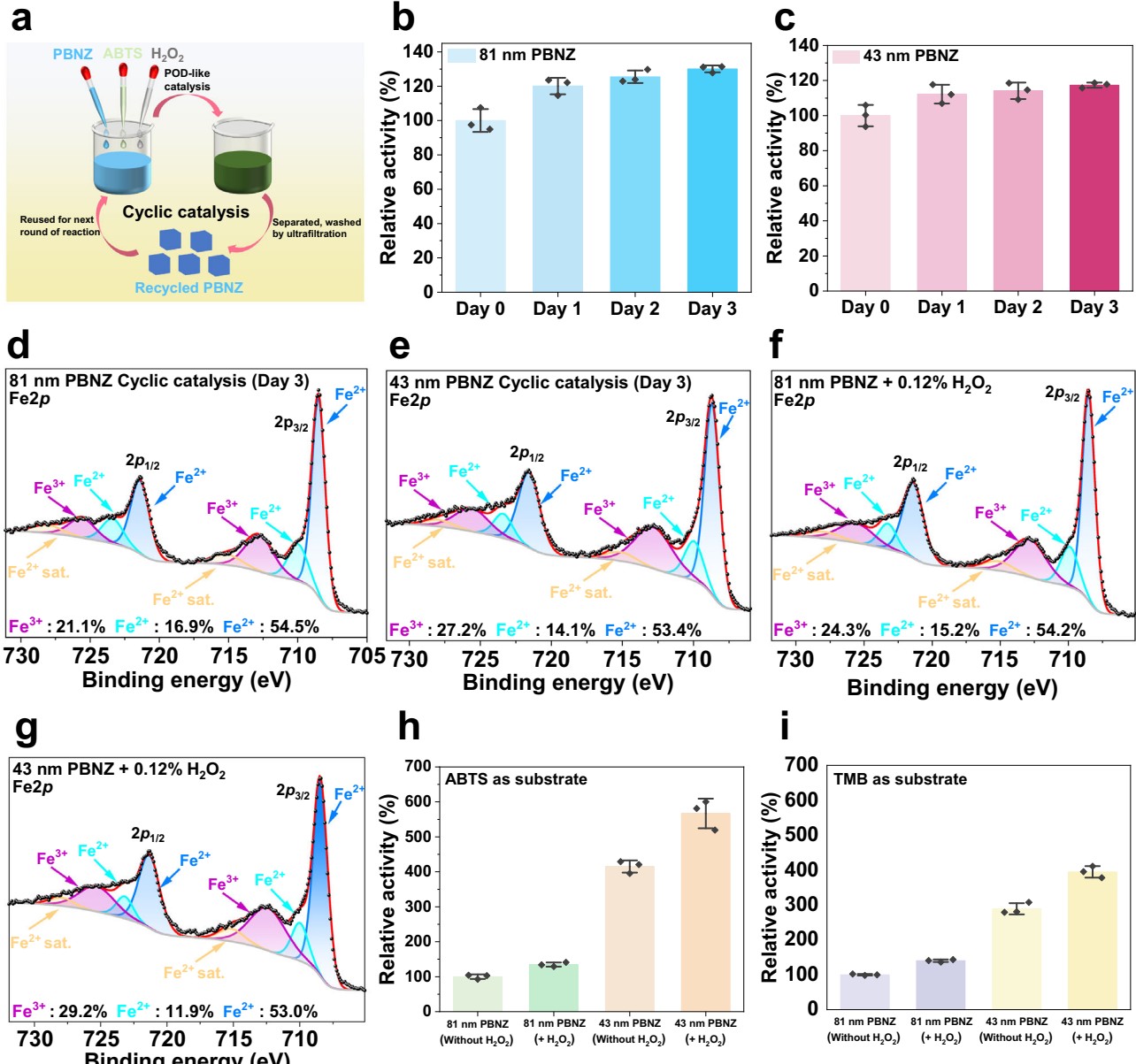

**Fig. 3 | Characterization of the PBNZ recycled from POD-like long-term catalysis. a** Illustration of the POD-like cyclic catalysis process. Relative POD-like activity of the **b** 81 nm and **c** 43 nm PBNZ recycled from cyclic catalysis (n = 3 independent experiments). Fitting XPS Fe2p spectrum of the **d** 81 nm and **e** 43 nm PBNZ recycled from cyclic catalysis. Fitting XPS Fe2p spectrum of the **f** 81 nm and **g** 43 nm PBNZ recycled from the incubation with 0.12% $H_2O_2$. Relative POD-like activity of the PBNZ recycled from the incubation with 0.12% $H_2O_2$ using **h** ABTS and **i** TMB as chromogenic substrate (n = 3 independent experiments). Error bars represent SD from three independent measurements and all data are presented as mean values ± SD.

together, the irreversible oxidation of PBNZ by $H_2O_2$ is responsible for their improved POD-like activity, and low-crystallinity PBNZ are more susceptible to oxidation. Although the addition of the reductive ABTS weakens this oxidation effect during actual catalysis, the increasing amount of surficial Fe (III) during POD-like catalysis remains significant.

**CAT-like long-term catalysis of PBNZ**
The catalytic behavior of PBNZ shifts from POD to CAT as the pH of the solution increases from acidic to neutral conditions[24]. In the prolonged CAT-like catalysis, the self-decomposition of $H_2O_2$ catalyzed by PBNZ was observed in pure water using a hydrogen peroxide detection kit. Due to the weaker oxidizing ability of $H_2O_2$ at neutral environment, the final concentration of $H_2O_2$ was increased to 3.6%, compared with 0.12% used in the POD-like experiments.

Pure water was chosen over phosphate-buffered saline (PBS) due to the long-term instability of PBNZ in PBS[24]. The detection kit uses molybdic acid, which chelates with $H_2O_2$ to form a yellow-colored product with an absorption peak at 405 nm. As shown in Supplementary Fig. 13, the absorbance at 405 nm of the reaction system continuously decreased, indicating that the PBNZ remained active as CAT mimetics for at least 24 h.

Similar to the prolonged POD-like catalysis, significant physicochemical changes were observed in the PBNZ recycled from the long-term CAT-like reactions. Aggregation was evident, as indicated by the increased hydrodynamic diameter of the PBNZ (Fig. 4a). The surface negative charge of the PBNZ decreased rapidly (Fig. 4b), and the color of the recycled PBNZ solution turned slightly green, particularly for the 43 nm PBNZ (Supplementary Fig. 14). This color

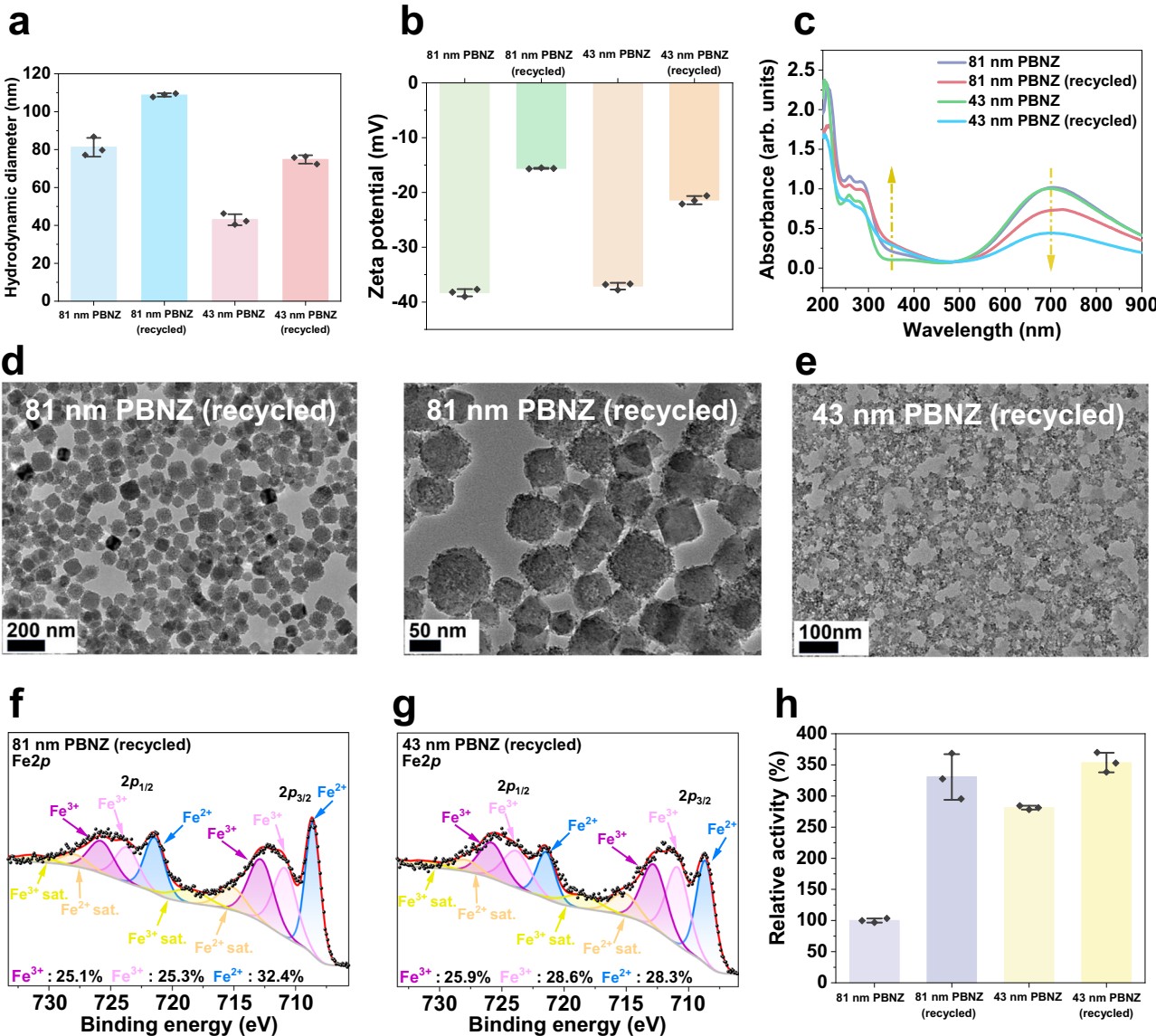

**Fig. 4 | Characterization of the PBNZ recycled from CAT-like long-term catalysis. a** Hydrodynamic diameter (n = 3 independent experiments), **b** zeta potential (n = 3 independent experiments), **c** UV-vis spectra, **d**, **e** TEM images, **f**, **g** fitting XPS Fe2*p* spectrum, and **h** relative CAT-like activity of the recycled PBNZ (n = 3 independent experiments). TEM images were collected three times with similar results. Error bars represent SD from three independent measurements and all data are presented as mean values ± SD.

change was accompanied by a decreased adsorption intensity around 700 nm and an increase in 300–500 nm range (Fig. 4c). Additionally, the surface of the 81 nm became slightly rough (Fig. 4d), indicating mild structural disruption similar to the deformation of PB or its analogs after $H_2O_2$ exposure as reported in previous[38,39]. XPS spectra revealed the disappearance of N-coordinated Fe (II) and a decrease of C-coordinated Fe (II). In contrast, the amount of N-coordinated Fe (III) increased and a new peak representing C-coordinated Fe (III) at 710.9 eV appeared (Fig. 4f, g). Additionally, oxygenated groups such as Fe-OH and Fe=O prominently emerged (Supplementary Fig. 15). Increased Fe leaching was also observed (Supplementary Fig. 16). These results collectively revealed the oxidative effects of $H_2O_2$ on PBNZ during CAT-like catalysis. Consequently, the CAT-like activity of the recycled PBNZ was notably enhanced (Fig. 4h). In conclusion, even in a neutral environment, the irreversible oxidation effect during CAT-like long-term catalysis is evident when $H_2O_2$ dosage is sufficiently high, contributing to the improvement of the CAT-like activity of the recycled PBNZ.

## Catalytic mechanism of PBNZ as POD and CAT mimetics

The active intermediates Fe-OH and Fe=O are considered crucial for the catalytic activity of PBNZ. To identify these intermediates, methyl phenyl sulfoxide (PMSO) was used as a probe. PMSO can be specifically oxidized by Fe-OH and Fe=O, with the latter process producing $PMSO_2$ which exhibits light adsorption at 215 nm. Thus, the formation of Fe=O during catalysis can be quantified by measuring the transformation efficiency ($\eta$) of $PMSO_2$ from PMSO[40]. As shown in Fig. 5a, both 81 nm and 43 nm PBNZ systems exhibited high $\eta$ values at pH 3.6, indicating significant Fe=O formation. In contrast, the 90 nm PBNZ system displayed a lower $\eta$ value (57.0 ± 2.7%) (Supplementary Fig. 17), highlighting the difficulty of oxidizing more crystalline 90 nm PBNZ. The $PMSO_2$ production and $\eta$ value decreased rapidly in pure water (pH 7.0), partly due to the reduced oxidizing ability of $H_2O_2$ at neutral pH, which directly lowers the production of Fe=O[24]. Additionally, the competitive CAT-like reaction at neutral pH consumes Fe=O to produce $O_2$ (Supplementary Fig. 18). The incomplete transformation of $PMSO_2$ also indicated the presence of Fe-OH during PBNZ catalysis,

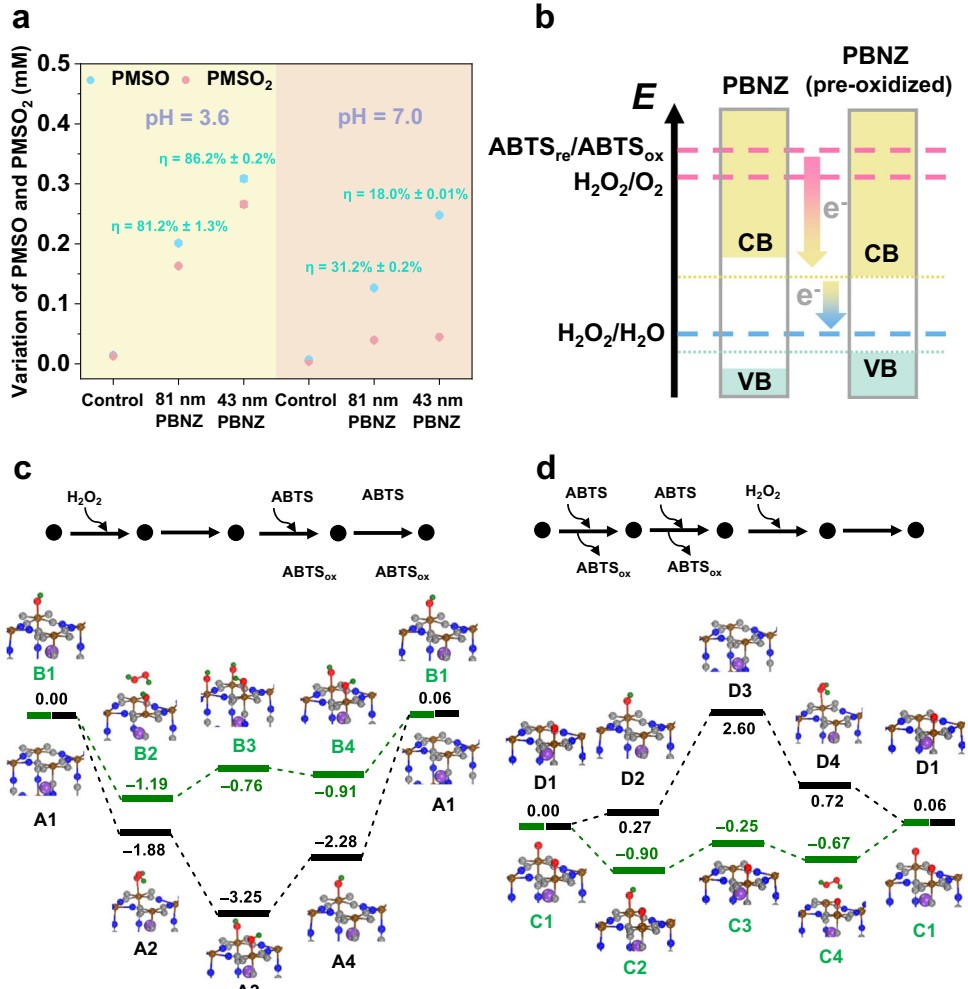

**Fig. 5 | Dual-path electron transfer mechanism of PBNZ. a** PMSO and $PMSO_2$ detection by HPLC (n = 3 independent experiments, error bars represent SD from three independent measurements and data are presented as mean values ± SD). **b** Energy levels of catalytic substrates and conduction/valence band of PBNZ (VB valence band, CB conduction band). **c** Mechanism and energy profiles (energy in eV) mediated by VBP. **d** Mechanism and energy profiles (energy in eV) mediated by CBP.

further confirmed by electron spin resonance spectrometer (EPR) (Supplementary Fig. 19a). The contribution of $^1O_2$ to PBNZ-mediated catalysis was excluded as no signal was detected by EPR (Supplementary Fig. 19b). In summary, both Fe-OH and Fe=O intermediates are involved in the catalysis of PBNZ, as demonstrated through various experimental analyses.

We focused on identifying the electron transfer pathway during the POD and CAT-like catalysis of PBNZ. Similar to our previous work predicting superoxide-dismutase activity through energy level principle[41], the electron transfer pathway of PBNZ could also be initially estimated from the perspective of energy level. As a semiconductor, the N-coordinated Fe and C-coordinated Fe in PB contribute to their conduction band and valence band, respectively[33]. The optical band gaps of the 81 nm and 43 nm PBNZ were obtained from their adsorption spectra (Supplementary Fig. 20). Their energy levels relative to the standard hydrogen electrode (SHE) were then calculated and compared with those of ABTS and $H_2O_2$ (Supplementary Table 1). The conduction band energy level ($E_{CB}$) of PBNZ lies between the energy levels of $H_2O_2/H_2O$ and ABTS/$ABTS_{ox}$, while their valence band energy level ($E_{VB}$) is below the energy level of $H_2O_2/H_2O$. Thus, the CBP mediated by the conduction band of PBNZ is estimated to be thermodynamically preferred during the POD and CAT-like

catalysis compared with the VBP. The catalysis begins with the electron injection from ABTS or $H_2O_2$ into the PBNZ conduction band, forming $ABTS_{ox}$ or $O_2$. Subsequently, electron transfer from the PBNZ conduction band to $H_2O_2$ generates $H_2O$ (Fig. 5b). Furthermore, the oxidation effect of $H_2O_2$ narrowed the band gap ($E_g$) of PBNZ with an increase of $E_{VB}$ (Supplementary Fig. 20, Fig. 5b). Notably, the oxidation primarily occurs on the surface of the particles, whereas the optical method measured the $E_g$ in the bulk phase. Thus, although the calculated $E_{VB}$ of the oxidized PBNZ remains below the $H_2O_2/H_2O$ energy level, the actual increase in the surficial $E_{VB}$ is likely higher and may elevate the valence band energy level above $H_2O_2/H_2O$, facilitating the occurrence of VBP.

Density functional theory (DFT) calculations were conducted to elucidate the atomistic-level mechanism underlying the POD-like activity of PBNZ and the role of oxidation in enhancing their activity. Based on the experimental results, Fe-OH and Fe=O are likely the key intermediates in the catalysis. Therefore, three PBNZ structures were evaluated: pristine PBNZ (A1 of Fig. 5c), PBNZ modified with -OH (B1), and PBNZ with =O (C1 and D1). For any given PBNZ structure, the catalysis may be initiated by reacting with $H_2O_2$ (via the VBP) or ABTS (via the CBP). The calculations revealed that the pristine PBNZ (A1) exhibited minimal POD-like activity. Instead, it could be easily oxidized

by $H_2O_2$ to form a thermally stable intermediate (A3), which had a deep potential energy well (−3.25 eV) and was difficult to be reduced back to A1 by ABTS (Fig. 5c). Further calculations indicated that the two-OH structure A3 could dehydrate to form a one-O structure (D1), with an energy change of 0.7 eV. As more OH groups are present on PBNZ, the dehydration becomes more energetically favorable. The dehydration of three-OH structure B3 to one-OH-and-one-O structure C2 was energetically favorable with an energy change of −0.11 eV. Thus, the concentration of Fe=O would increase with the extent of oxidation. These findings align with experimental observations showing that unoxidized PBNZ had lower POD-like activity and that Fe-OH and Fe=O groups were prevalent in the PBNZ structures due to irreversible oxidation.

In contrast, the oxidized PBNZ structure B1 had a shallow potential energy well and should have considerable POD-like activity. As illustrated in Fig. 5c, the one-OH structure B1, representing the moderately oxidized PBNZ, could catalyze the reaction between $H_2O_2$ and ABTS via the VBP. Initially, an $H_2O_2$ molecule dissociated on B1 to form B3, resulting in an energy decrease of −0.76 eV. B3 was then reduced back to B1 by two ABTS molecules, completing the catalytic cycle. The two-O structure C1, representing the heavily oxidized PBNZ, could catalyze the POD-like reaction via the CPB. C1 was first reduced by two ABTS to form one-O structure C3, with an energy decrease of −0.25 eV. C3 was then oxidized back to C1 by $H_2O_2$ (Fig. 5d). One-O structure D1, however, could not undergo the POD-like catalysis via the CBP due to the high energy of the intermediate D3 (2.6 eV). Similarly, the two-OH structure E1 as shown in Supplementary Fig. 23 could catalyze the reaction via the VBP, the three-OH structure B3 could catalyze the reaction via the CBP (B3-B4-B1-B2-B3), and the one-O structure could catalyze the reaction via the VBP (C3-C4-C1-C2-C3). Notably, the Fe=O is present in both the PBNZ and horseradish peroxidase (HRP)-mediated POD-like processes, involving two sequential one-electron transfers from the chromogenic substrates, which to some extent indicates similarity. However, the specific transformation of Fe=O during their catalysis differs (PBNZ: Fe=O → Fe-OH → Fe; HRP: Fe=O+ → Fe=O → Fe[42]). Moreover, since the CAT-like process can be viewed as a special POD-like process where ABTS is replaced by reductive $H_2O_2$ under neutral pH, the above conclusions are also applicable to explain the enhancement of CAT-like activity in oxidized PBNZ. Taken together, unoxidized PBNZ (e.g., A1) exhibits little catalytic activity. However, once dispersed in the catalytic solution and irreversibly pre-oxidized by $H_2O_2$, moderately and heavily oxidized PBNZ show significant POD and CAT-like activity via the VBP and CBP, respectively.

In summary, the catalytic mechanism of PBNZ has been elucidated through the experimental and theoretical analyses on the PBNZ recycled from long-term catalysis. Our results demonstrate the unique advantage of PBNZ, which exhibits self-enhancing POD and CAT-like activities, compared with natural enzymes and iron-based nanozymes that deplete over time. The pre-oxidation effect of $H_2O_2$ leads to an increase in the valence state of surface Fe, accompanied by the formation of oxygenated groups, which is crucial for enhancing both VBP and CBP during catalysis. The proposed dual-path electron transfer mechanism thoroughly explains the catalytic behavior of PBNZ, ensuring a long service life.

## Methods
### Chemicals
All chemicals were of analytical grade and were used without further purification. Iron (III) chloride hexahydrate ($FeCl_3 \cdot 6H_2O$), Citric acid monohydrate ($C_6H_8O_7 \cdot H_2O$), Sodium acetate ($C_2H_3NaO_2$), Acetic acid ($CH_3COOH$, 98.0%), Hydrogen peroxide ($H_2O_2$, 30%), Dimethyl sulfoxide (DMSO), Acetonitrile were all obtained from Sinopharm Chemical Reagent Co., Ltd. Potassium hexacyanoferrate (II) trihydrate ($K_4[Fe(CN)_6] \cdot 3H_2O$) were purchased from Shanghai Lingfeng Chemical

reagent Co., Ltd. TMB and ABTS were bought from Aladdin (Shanghai, China). PMSO and $PMSO_2$ were purchased from Shanghai Yuanye Bio-Technology Co., Ltd. 2,2,6,6-tetramethyl-4-piperidine (TMP) and 5,5-dimethyl-1-pyrroline-N-oxide (DMPO) were bought from Sigma Aldrich. Hydrogen peroxide detection kit were obtained from Jiangsu KeyGEN BioTECH Co., Ltd. Deionized water was used in all experiments.

### Characterization
Particle size, morphology, crystallinity, and element component analysis of PBNZ were conducted using TEM (JEM-2100, JEOL, Japan) equipped with EDS and Digital Micrograph software (v3.2). Hydrodynamic diameter and zeta potential were measured by DLS (Nano-ZS90, Malvern, England) coupled with Malvern Zetasizer software (v7.12). UV-vis absorption spectra were pictured on a UV-VIS-NIR spectrophotometer (UV-3600, Shimadzu, Japan) coupled with UV Probe software (v2.42). Fe element concentration in filtrate solution was determined by an inductively coupled plasma spectrometer (ICP)-mass spectrum (MS) device (ICP-MS 7800, Agilent, USA) coupled with ICP Expert II software. The valence state of elements on the PBNZ surface was measured by XPS (Escalab 250Xi, Thermo, USA) coupled with Thermo Advantage software (v5.967).

### Synthesis of PBNZ by double injection method
PBNZ with different sizes and crystallinity were prepared by our previously reported double injection method[35]. In a general procedure, 1 mmol $FeCl_3$ and 3 mmol $C_6H_8O_7$ were added into 20 mL deionized water under magnetic stirring for 30 min to form $Fe^{3+}$-citric acid solution. 1 mmol $K_4[Fe(CN)_6]$ was added into 20 mL deionized water under magnetic stirring for 10 min. Prior to the double injection, $x$ mL $Fe^{3+}$-citric acid was added into the reaction flask containing 60 mL deionized water under vigorous stirring at 60 °C. Subsequently, (20 - $x$) mL $Fe^{3+}$-citric acid and 20 mL $K_4[Fe(CN)_6]$ solution were simultaneously injected into the flask at a flow rate of 40 mL $h^{-1}$. The reaction solution was continuously stirred with a certain reaction time after the injection. The $x$ value was set as 1.0 mL and 20.0 mL to obtain 81 nm and 43 nm PBNZ with low crystallinity, respectively. The reaction time was set as 25 min and 2 h for 81 nm and 43 nm PBNZ, respectively.

### Measurement of the specific catalytic activity of PBNZ as POD mimetics
Typically, 200 μL HAc-NaAc buffer (0.2 M, pH = 3.6), 20 μL PBNZ, 10 μL 10 mg $mL^{-1}$ chromogenic substrate (ABTS, TMB) and 30 μL 1% $H_2O_2$ were sequentially added into the reaction system at room temperature. The absorbance change of the reaction system which was monitored in 60 s by a Microplate Reader (Model 680, BIO-RAD, USA) (coupled with Tecan i-control software (v1.6.19.2)) displayed the POD-like activity of PBNZ. When ABTS was selected as substrate, the concentration of PBNZ was set as 60 μg $mL^{-1}$, absorbance change of solution was observed at 415 nm; for TMB, the concentration of PBNZ was diluted to 6 μg $mL^{-1}$, absorbance change of the solution was observed at 650 nm.

The $a_{POD}$ of PBNZ as POD mimetics was defined following Eq. (1) based on our previous work[36]:

$$a_{POD} = [V(\Delta A/\Delta t)]/(\varepsilon l m_{Fe}) \tag{1}$$

where $a_{POD}$ is the specific catalytic activity of PBNZ (U $mg^{-1}$); $V$ is the total volume of the reaction solution (μL); $\Delta A/\Delta t$ is the initial change rate of reaction solution absorbance after correcting with reagent blank rate ($min^{-1}$); $\varepsilon$ is the molar absorption coefficient of chromogenic substrate derivative (ABTS: 36000 $mol^{-1}$ L $cm^{-1}$; TMB: 39000 $mol^{-1}$ L $cm^{-1}$); $l$ is the optical path of the cuvette (cm); $m_{Fe}$ is the total Fe element mass contained in added PBNZ (mg).

## Measurement of the specific catalytic activity of PBNZ as CAT mimetics

The CAT-like activity of PBNZ was measured in pure water. Typically, 5 mL pure water and 50 μL 80 μg mL$^{-1}$ PBNZ were mixed in the measurement system at room temperature. 167 μL 30% $H_2O_2$ was added into the reaction system and the concentration change of dissolved oxygen (DO) in 5 min which was monitored by a dissolved oxygen meter (JPSJ-605F, Lei ci, China), showing the CAT-like activity of PBNZ. Measurement of reagent blank rate: all the procedures are the same as described above except that PBNZ was replaced by deionized water.

The $a_{CAT}$ of PBNZ as CAT mimetics was defined following Eq. (2):

$$a_{CAT} = [2V(\Delta D / \Delta t)]/m_{Fe} \tag{2}$$

where $a_{CAT}$ is the specific catalytic activity of PBNZ (U mg$^{-1}$); $V$ is the total volume of the reaction solution (mL); $\Delta D/\Delta t$ is the initial change rate of the dissolved oxygen in the reaction solution after correcting with reagent blank rate (mmol L$^{-1}$ min$^{-1}$); $m_{Fe}$ is the total Fe element mass contained in added PBNZ (mg).

## Long-term catalysis of PBNZ

For POD-like activity, 3.2 L HAc-NaAc buffer, 320 mL 60 μg mL$^{-1}$ PBNZ, 160 mL 10 mg mL$^{-1}$ ABTS and 480 mL 1% $H_2O_2$ solution were mixed in the reaction system (final concentration of reagents: PBNZ: 4.62 μg mL$^{-1}$; ABTS: 0.385 mg mL$^{-1}$; $H_2O_2$: 0.12%). The catalytic reaction lasted for 24 h at room temperature under magnetic stirring. After the reaction, PBNZ was separated and washed with deionized water several times by ultrafiltration following with the reuse of the as-collected PBNZ for a new round of catalysis. Ultimately, three rounds of the above cyclic catalysis were conducted. PBNZ from different cyclic days were recycled for further characterization.

For CAT-like activity, 2 L pure water, 200 mL 60 μg mL$^{-1}$ PBNZ, and 300 mL 30% $H_2O_2$ solution were mixed in the reaction system (final concentration of reagents: PBNZ: 4.80 μg mL$^{-1}$; $H_2O_2$: 3.60%). The catalytic reaction lasted for 24 h at room temperature under magnetic stirring. After the reaction, PBNZ was separated and washed with deionized water several times by ultrafiltration. PBNZ were recycled for further characterization.

## Free radical detection by EPR

For the detection of·OH, DMPO was used as the spin-trapping agent to trap ·OH. After the addition of PBNZ, $H_2O_2$ and DMPO into the HAc-NaAc buffer, solutions were monitored by EPR (EMX PLUS, Bruker, Germany) coupled with Bruker Xenon software (v1.2). The concentration of PBNZ, $H_2O_2$ and DMPO were 4.62 μg mL$^{-1}$, 0.12% and 50 mM, respectively.

For the detection of $^1O_2$, DMPO was substituted by TMP as the spin-trapping agent. The concentration of reagents is the same as ·OH detection experiment.

## High valent Fe detection by HPLC

3 mL HAc-NaAc buffer or deionized water, 300 μL 60 μg mL$^{-1}$ PBNZ, 150 μL 10 mM PMSO and 450 μL 1% $H_2O_2$ were mixed to evaluate the producibility of Fe in high valence when PBNZ were mixed with $H_2O_2$. After 24 h incubation, PBNZ were removed by ultrafiltration. Afterward, PMSO and PMSO$_2$ in the filtrate were examined by HPLC (HPLC e2695, Waters, USA) coupled with Empower 3 software according to the reported article[40]. Briefly, solutions were separated by Waters C18 column (250 × 4.6 mm, 5 μm particle size, XBridge). The mobile phase consisted of 30% acetonitrile and 70% water with 0.1% acetic acid at a flow rate of 1.0 mL min$^{-1}$. PMSO and PMSO$_2$ were detected at wavelengths of 230 nm and 215 nm, respectively. The transformation

efficiency $\eta$ was defined following Eq. (3):

$$\eta = \Delta C_{PMSO_2}/\Delta C_{PMSO} \tag{3}$$

where $\Delta C_{PMSO}$ and $\Delta C_{PMSO_2}$ are the consumption amount of PMSO and formation amount of PMSO$_2$, respectively.

## Energy level calculation of PBNZ versus SHE

$E_{CB}$ and $E_{VB}$ of PBNZ versus SHE were calculated using the following empirical formula (Eqs. (4) and (5))[43]:

$$E_{CB} = X - 4.5 - 0.5E_g \tag{4}$$

$$E_{VB} = X - 4.5 + 0.5E_g \tag{5}$$

where $E_{CB}$ and $E_{VB}$ are the conduction band energy level and valence band energy level of PBNZ respectively (eV). $X$ represents the electronegativity of PBNZ ($Fe_4[Fe(CN)_6]_3$) (Eq. (6)):

$$X = [\chi(Fe)^7 \chi(C)^{18} \chi(N)^{18}]^{1/43} \tag{6}$$

where $\chi$ which reflects the electronegativity of atoms for Fe, C and N are 4.03, 6.26 and 7.30, respectively.

$E_g$ was calculated according to Eq. (7):

$$(\alpha h\nu)^n = B(h\nu - E_g) \tag{7}$$

in which $E_g$, $\alpha$, $h\nu$ and $B$ are optical band gap (eV), absorption coefficient, photon energy (J) and proportionality constant, respectively. $n$ represents the transition categories of semiconductor ($n = 1/2$ for indirect transition; $n = 2$ for direct transition). Given that PB is considered as indirect transition semiconductor[44] and $\alpha$ is linear with absorbance ($A$), variation of $(Ah\nu)^{0.5}$ versus $h\nu$ were plotted. The linear range of these plots was extrapolated to the x-axis ($y = 0$) to obtain the value of $E_g$.

## Structural models

According to the experimental cell parameters of PBNZ, an ideal structural model of PBNZ was established. The molecular formula of the model was $K_4Fe_8(CN)_{24}$ and it had $F43m$ symmetry. Firstly, the cell parameters were optimized and the magnetic moments of Fe atoms were calculated. The results showed that the N- and C- coordinated Fe were attributed to $Fe^{3+}$ and $Fe^{2+}$, respectively. Then, the PBNZ (001) slab was cut from the cell structure for the calculation of the subsequent catalytic reaction. In order to save computing costs, the bottom four atomic layers were fixed during the optimized process. More details of the calculations can be found in the supplementary computational details (Supplementary Figs. 21 and 22).

## Computational methods

The spin polarized density functional theory calculations were applied to investigate reaction process. The geometry optimization of all structures was performed using the projector augmented wave (PAW) method[45] implemented in the Vienna Ab initio Simulation Package (VASP, version 5.4.4)[46]. The cutoff energy of plane-wave was set to 400 eV. The Perdew–Burke–Ernzerhof (PBE) exchange-correlation functional with the generalized gradient approach (GGA) was applied. The $U_{eff}$ value was set to 6.0 because the band gap calculated using the $U_{eff}$ value of 6.0 was close to the experimental band gap[47]. All slab structures have a vacuum value of 15 Å in $z$ direction to avoid interaction between adjacent cells. The electronic and geometry optimization convergence criteria were set to 10$^{-5}$ eV and 0.02 eV Å$^{-1}$, respectively. Monkhorst-Pack k point mesh was set to 5 × 5 × 5 when optimizing the cell parameters of $K_4Fe_8(CN)_{24}$. However, Monkhorst-

Pack k point mesh was set to $3 \times 3 \times 1$ for calculating other slab structures. All small molecules, such as $H_2O_2$, $H_2O$, $\cdot OH$, ABTS and ABTS-H were placed in a cube box of $15 \times 15 \times 15$ Å, optimizing geometry by using k point of $1 \times 1 \times 1$.

## Reporting summary

Further information on research design is available in the Nature Portfolio Reporting Summary linked to this article.

## Data availability

The experimental and computational data generated in this study are provided in Source Data file. All data underlying this study are available from the corresponding author upon request. Source data are provided with this paper.

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

## Acknowledgements

Y.Z. acknowledges the financial support for this work from the National Key Research and Development Program of China (2022YFA1205802, 2022YFC2406504), National Natural Science Foundation of China (82072067, 61821002), General Project of Jiangsu provincial Health Commission (M2022061), Scientific research foundation of Nanjing Health Commission (YYKK20232), Natural Science Foundation of Jiangsu Province (BK20211169), Jiangsu Planned Projects for Postdoctoral Research Funds (2021K518C) and the Fundamental Research Funds for the Central Universities (2242020k10017).

## Author contributions

Y.Z. and X.G. conceived and supervised the project; K.F. performed experiments and prepared manuscript; Z.W. performed the DFT calculations and prepared manuscript; S.W., G.W. and H.D. helped with data analysis; H.H., H.W. and M.M. assisted with article revisions.

## Competing interests

The authors declare no competing interests.
