## [Peer Review File · Nature Communications]

Reviewers' Comments:

Reviewer #1:

Remarks to the Author:

The authors investigated the pathways of enzyme-like activity (POD-like and CAT-like) of Prussian blue nanozymes (PBNZ). They found a dual-path electron transfer mechanism for both POD and CAT mimetics, as well as "self-increased" catalytic activity due to the oxidation of Fe centres to Fe(III) at the surface.

I am an expert in Prussian blue materials but not in nanozyme research (my field is solid-state chemistry). I therefore cannot properly judge the novelty and importance of the work.

The materials, as well as the mechanisms, are characterized in detail. The results seem reproducible and support the drawn conclusions.

As for the presentation of the work: The writing is not concise, hard to read, and overall the text is very lengthy. Also, the figures are not well presented and difficult to decipher. Overall, the manuscript is rather hard to read.

I would therefore recommend either major revisions, as the overall text and figures need to be changed to meet the standards of Nature Communications, or transfer to a lower impact factor and more specialized journal, for which the detailed descriptions may be adequate.

I would happily review the revised version of the article again.

My comments:

* The characterization is very lengthy and should go to a large extent to the SI. As far as I understood, the synthesis of the materials was previously reported and therefore it is not new.

* The English is hard to read. In particular, the introduction was difficult to understand. Frequent errors: the 3rd person singular "s" of the conjugated verb is often forgotten; articles (the, a) are often forgotten; plural "s" is sometimes forgotten; English expressions often are wrong. I would recommend passing the article to a native English speaker for correction before submitting it again to Nat. Commun. or a different journal.

* Many graphs within one figure are too small, which makes the writing unreadable, and hard to recognize anything. In many cases, one Figure contains too many graphs or subfigures. For example, Figure 5c is divided again into 4 panels. A lot of the graphs within one Figure could go to the SI as they are not crucial for the storyline.

* From my understanding, Berlin green and Prussian yellow are the same (oxidized) compound. This does not become clear in the introduction and is somewhat confusing.

* Since the double injection method used here seems vital in this work, it would be good to briefly describe it in one or two sentences.

* On page 6, the authors claim that the Fe:N ratio is lower than what is predicted by stoichiometry. This, in turn, would mean that there is some N (or CN), which is not bound to Fe, isn't it? What could be the reason?

* On page 6, the authors speculate that the combination of Fe³⁺ and [Fe(CN)₆]⁴⁻ is not adequate. What is meant by "not adequate" here?

*On page 10, what is meant by "magnifying the faint changes of nanozymes"?

* It is not entirely clear if leaching of Fe is beneficial or detrimental. It seems like both are claimed but maybe I misunderstood.

* If there is substantial leaching, how can be made sure that the catalytic process occurs on the PBNZ nanoparticle and is not catalyzed by Fe ions in solution as well?

* The reported decrease in bandgap upon H₂O₂ oxidation is only 0.01 eV, which is within the error of the fitting.

Reviewer #2:

Remarks to the Author:

The manuscript entitled 'Elucidating the catalytic mechanism of Prussian blue nanozymes with self-increasable activity from valence state and energy level perspectives' reported the self-enhancing activity of Prussian blue nanozymes (PBNZ) acting as peroxidase (POD) and catalase (CAT) mimetics during prolonged catalysis. The unclear catalytic process of PBNZ was explored from the perspectives of valence state and energy level, supported by designed experimental evidence. The proposed dual-path electron transfer process contributed to a valuable understanding on the catalytic mechanism of nanozymes. The manuscript can be considered to be published after addressing the following concerns.

1. I recommend emphasizing the catalytic mechanism of the dual-path electron transfer process throughout the entire manuscript. The current version lacks focus. Additionally, it is advisable to include more results related to the catalytic mechanism, such as model construction and simulation of the catalytic process.

2. In Fig. 3b, the TEM image suggests the presence of aggregated 43 nm PBNZ. How does this aggregation potentially affect their catalytic behavior?

3. The authors noted an increased leakage of Fe in Fig. 7 and Fig. 9 following the incubation of PBNZ with H₂O₂. Please discuss the contribution of Fe leaching to POD- and CAT-like catalysis?

4. When discussing the Fe (IV) process in PBNZ catalysis, it is recommended to describe the mechanism of horseradish peroxidase, which can underscore the similarities between PBNZ and natural enzymes.

5. The energy band gap narrowing of PBNZ after oxidation was relatively weak. A more detailed explanation of this phenomenon should be provided.

6. In the Fe (IV) detection using 90 PBNZ (Fig. 8a), the transformation efficiency of PMSO/PMSO₂ was quite low and further decreased during the detection in neutral pH (Fig. 10a). The reasons underlying this observation need to be explained.

7. The main manuscript includes an abundance of figures. The dual-path electron catalytic mechanism and the supported figures needed to be highlighted. Other figures are suggested to move to the supplementary information.

8. The whole manuscript should undergo proofreading, and any grammar mistakes should be corrected.

Reviewer #3:

Remarks to the Author:

The authors present an interesting and original study about the catalytic activity of Prussian blue nanozymes from a mechanistic approach. The manuscript is interesting in the field but there are some parts difficult to follow mainly due to inappropriate use of the language.

Some examples are:

Lines 344-345: Given that Fe (II) dominated the surface, their surface state was more like PW.

Line 381. The Apparent physicochemical properties change.... Refer to the changes described in the previous paragraph?

Lines 386-387: Color visibly changed, how? Decreased characteristic absorption at 700 nm, means that the maximum is displaced or that there is a decrease in the absorption intensity?

Conclusion's section is not clear. Authors conclude in lines 364, 421 and 444 (probably the real conclusion part), corresponding to the last part of each section.

Sentence of lines 444 to 446 is not clear.

Line 453, what is the meaning of The above findings are instructive for life span judgement?

Minor remarks:

All over the text nanozymes tested are of 81 and 43 nm but in the method section, line 478, describes the synthesis of 90 and 43 nm.

Figure 3b) should show the size as in figure 2b.

Line 139, result should be omitted.

Figure 4. Colors of the lines for 43 and 81 nm are confusing.

Dear reviewers,

We sincerely thank you for your constructive comments. We have revised and improved our manuscript according to your suggestions. Our detailed, point-by-point responses are listed in the following pages. A copy tracking all the changes made in our revision has also been submitted. We hope that our responses could be acceptable.

Sincerely yours,

Yu Zhang

Southeast University, Nanjing 211102, P. R. China

E-mail address: zhangyu@seu.edu.cn

Point-by-point responses to the reviewers' comments:

Reviewer #1 (Remarks to the Author):

The authors investigated the pathways of enzyme-like activity (POD-like and CAT-like) of Prussian blue nanozymes (PBNZ). They found a dual-path electron transfer mechanism for both POD and CAT mimetics, as well as “self-increased” catalytic activity due to the oxidation of Fe centres to Fe(III) at the surface.

I am an expert in Prussian blue materials but not in nanozyme research (my field is

solid-state chemistry). I therefore cannot properly judge the novelty and importance of the work.

The materials, as well as the mechanisms, are characterized in detail. The results seem reproducible and support the drawn conclusions.

As for the presentation of the work: The writing is not concise, hard to read, and overall the text is very lengthy. Also, the figures are not well presented and difficult to decipher.

Overall, the manuscript is rather hard to read.

I would therefore recommend either major revisions, as the overall text and figures need to be changed to meet the standards of Nature Communications, or transfer to a lower impact factor and more specialized journal, for which the detailed descriptions may be adequate.

I would happily review the revised version of the article again.

Response. Thank you very much for your valuable suggestions on our work. We have revised our manuscript carefully by following the guidance that you provided. Detailed revisions are shown in the following responses.

Comment 1. The characterization is very lengthy and should go to a large extent to the SI. As far as I understood, the synthesis of the materials was previously reported and

therefore it is not new.

Response. We appreciate the reviewer's suggestion. To make the main manuscript more concise and readable, the basic characterizations of the as-synthesized PBNZ were transferred to SI (renumbered as Supplementary Fig. 1 and Supplementary Fig. 2) with supplementary discussions in the revised supplementary information.

Comment 2. The English is hard to read. In particular, the introduction was difficult to understand. Frequent errors: the 3rd person singular "s" of the conjugated verb is often forgotten; articles (the, a) are often forgotten; plural "s" is sometimes forgotten; English expressions often are wrong. I would recommend passing the article to a native English speaker for correction before submitting it again to Nat. Commun. or a different journal.

Response. We are extremely grateful to reviewer for pointing out this problem. To improve the English of our manuscript, the grammar and expression mistakes in the introduction section and other sections of the entire article were carefully corrected and highlighted in blue in the revised manuscript.

Comment 3. Many graphs within one figure are too small, which makes the writing unreadable, and hard to recognize anything. In many cases, one Figure contains too many graphs or subfigures. For example, Figure 5c is divided again into 4 panels. A lot of the graphs within one Figure could go to the SI as they are not crucial for the storyline.

Response. We are grateful for this suggestion. We have greatly improved the quality of the figures in our manuscript. Figures that are not crucial for the storyline in the main

manuscript were transferred to the SI and renumbered as Supplementary Fig. 1, Fig.2, Fig. 6, Fig. 7, Fig. 10, Fig. 12, Fig. 13, Fig. 14, Fig. 16, Fig. 17, Fig. 18 and Fig. 20 with supplementary discussions. The number of figures in the main manuscript was reduced to five with magnified size and reduced subfigures (Fig. 1-5).

Comment 4. From my understanding, Berlin green and Prussian yellow are the same (oxidized) compound. This does not become clear in the introduction and is somewhat confusing.

Response. Thank you very much for the comment. According to the relevant studies (*Advanced Functional Materials* 14 (2004) 224-232, *Thin Solid Films* 789 (2024) 140192), both of Berlin green (BG) and Prussian yellow (PY) are the oxidized compound of PB. Differently, PY is a yellow colored, fully oxidized product of PB in the chemical formula of $\text{Fe(III)[Fe(III)(CN)}_6]$. While BG is a green colored, partially oxidized product of PB which could be also regarded as a mixture of PB and PY in the ratio of 1:2 ($\text{KFe(III)}_3[\text{Fe(III)(CN)}_6]_2[\text{Fe(II)(CN)}_6]$). To clarify their difference, their chemical formulas were given and highlighted in red as shown in the introduction section of the revised manuscript.

Page 3, line 56-60: “For example, our initial work preliminarily speculated that PBNZ work as POD and CAT mimetics through a mechanism of Prussian blue (PB, $\text{KFe(III)[Fe(II)(CN)}_6]$) / Berlin green (BG, $\text{KFe(III)}_3[\text{Fe(III)(CN)}_6]_2[\text{Fe(II)(CN)}_6]$) / Prussian yellow (PY, $\text{Fe(III)[Fe(III)(CN)}_6]$) conversion with redox potential matching.²⁴”

Comment 5. Since the double injection method used here seems vital in this work, it would be good to briefly describe it in one or two sentences.

Response. Thank you very much for the suggestion. The double injection method was developed in our previous work for the size and crystallinity-controlled synthesis of PBNZ. A brief description of this method was added into the revised manuscript and highlighted in red.

Page 5, line 95-99: “PBNZ used for long-term catalysis in this work were prepared by our previously reported double injection method. By tuning the prior addition volume of Fe³⁺-citric acid solution (i.e., x value) and the reaction time, the size and the crystallinity of PBNZ could be facilely regulated.³⁵”

Comment 6. On page 6, the authors claim that the Fe:N ratio is lower than what is predicted by stoichiometry. This, in turn, would mean that there is some N (or CN), which is not bound to Fe, isn't it? What could be the reason?

Response. Thank you sincerely for your insightful question. Generally, the chemical formulas of ideal PB are KFe(III)[Fe(II)(CN)₆] (soluble PB) and Fe(III)₄[Fe(II)(CN)₆]₃ (insoluble PB) with a Fe:N ratio of 1:3 and 1:2.6, respectively. However, the element analysis of the as-prepared PBNZ by EDS method showed a lower Fe:N ratio (Fig. 2c, 2f). Moreover, the decreased Fe:N ratio was further confirmed by XPS data which reflected the surficial Fe:N ratio of PBNZ (Fig. 2a, 2d). The lower Fe:N ratio suggested the Fe missing in the actual structure of the synthesized PBNZ. It is speculated that C atoms are fully bonded to Fe due to the high stability of [Fe(CN)]⁴⁻ and the missing of

$[\text{Fe}(\text{CN})]^{4-}$ would increase the Fe:N ratio, it is thus inferred that there are some N atoms which are not bound to Fe^{3+} , resulting in the low Fe:N ratio.

The inner reason for this phenomenon is related to the synthesis process. During the synthetic process, the low solubility product constant (K_{sp}) of PB (3.0×10^{-41}) led to an ultra-fast formation process of nanoparticles, during which the combination of the Fe^{3+} and $[\text{Fe}(\text{CN})]^{4-}$ could not be fully conducted. To prove our hypothesis, the prepared PBNZ were immediately removed from the unpurified PBNZ solution by ultrafiltration once the synthesis was finished, the filtrates were analyzed to determine the residual components after the synthesis. As shown in Supplementary Fig. 4, the color of the obtained filtrates was visibly similar to Fe^{3+} in yellow (Supplementary Fig. 4a). Moreover, a slight adsorption around 300 nm was observed in the filtrates which further verified that the main component in the filtrates was Fe^{3+} (Supplementary Fig. 4b). Thus, it can be concluded that the filtrate is mainly composed of Fe^{3+} , demonstrating the lower usage of Fe^{3+} during the synthesis and missing Fe^{3+} in the structure of PBNZ. The above discussion was highlighted in red in the revised manuscript and the supplementary information.

Fig. 2 Surface composition and catalytic activity of prepared PBNZ. **a** XPS survey, **b** Fe 2p spectrum and **c** EDS spectrum of 81 nm PBNZ. **d** XPS survey, **e** Fe 2p spectrum and **f** EDS spectrum of 43 nm PBNZ. POD-like activity measurement using **g** ABTS and **h** TMB as substrates. **i** CAT-like activity measurement. Error bars represent standard deviation from three independent measurements.

Page 6, line 116-123: “Besides, the Fe:N ratio of 81 nm and 43 nm PBNZ detected by EDS and XPS analyses were apparently lower than their theoretical formula ($\text{KFe(III)[Fe(II)(CN)}_6]$ (Fe:N = 1:3) or $\text{Fe(III)}_4[\text{Fe(II)(CN)}_6]_3$ (Fe:N = 1:2.6) (Fig. 2a, c, d, f)), indicating the missing of Fe^{3+} in the bulk phase and surface of the as-prepared PBNZ due to the inadequate combination of reagents (i.e., the Fe^{3+} cannot fully react

with the $[\text{Fe}(\text{CN})_6]^{4-}$ during the ultra-fast synthesis). The insufficient usage of Fe^{3+} was further demonstrated by analyzing the components of the filtrate solution of the unpurified PBNZ (Supplementary Fig. 4).”

Supplementary Fig. 4 Composition detection on the filtrate solution of unpurified PBNZ. **a** Photos and **b** adsorption spectra of $[\text{Fe}(\text{CN})_6]^{4-}$, Fe^{3+} and filtrate solutions obtained from unpurified PBNZ. **c** Fe element detection on the filtrate solutions by ICP-MS. Error bars represent the standard deviation of ICP-MS measurement. Error bars represent standard deviation from three independent measurements.

Supplementary discussion for Supplementary Fig. 4: The unpurified PBNZ solution was ultrafiltered once the synthesis was finished and the color of the obtained filtrates was visibly similar to Fe^{3+} in yellow (Supplementary Fig. 4a). Moreover, a slight adsorption of the filtrate solutions around 300 nm was observed which further verified that the main component in the filtrate solutions was Fe^{3+} , leading to the missing of Fe^{3+} in the actual structure of prepared PBNZ (Supplementary Fig. 4b). The concentration of the unreacted Fe^{3+} during the synthesis process was determined by ICP-MS, which accounted for 6.5% and 4.8% of the total injected Fe^{3+} for the preparation of 81 nm and 43 nm PBNZ, respectively.

Comment 7. On page 6, the authors speculate that the combination of Fe^{3+} and $[\text{Fe}(\text{CN})_6]^{4-}$ is not adequate. What is meant by “not adequate” here?

Response. We appreciate the reviewer’s comment. It is known that the formation rate of PBNZ is very fast due to the low K_{sp} of PB ($K_{sp} = 3.0 \times 10^{-41}$). Therefore, it is logical to speculate that the combination of Fe^{3+} and $[\text{Fe}(\text{CN})_6]^{4-}$ is not adequate. In other words, Fe^{3+} cannot fully react with $[\text{Fe}(\text{CN})_6]^{4-}$ during the fast synthetic process, resulting in the Fe missing and the low Fe:N ratio of the obtained PBNZ (The Fe^{3+} missing was further proved as described in the response for comment 6.). The revised expression was highlighted in red in the revised manuscript.

Page 6, line 116-121: “Besides, the Fe:N ratio of 81 nm and 43 nm PBNZ detected by EDS and XPS analyses were apparently lower than their theoretical formula ($\text{KFe}(\text{III})[\text{Fe}(\text{II})(\text{CN})_6]$ (Fe:N = 1:3) or $\text{Fe}(\text{III})_4[\text{Fe}(\text{II})(\text{CN})_6]_3$ (Fe:N = 1:2.6) (Fig. 2a, c, d, f)), indicating the missing of Fe^{3+} in the bulk phase and surface of the as-prepared PBNZ due to the inadequate combination of reagents (i.e., the Fe^{3+} cannot fully react with the $[\text{Fe}(\text{CN})_6]^{4-}$ during the ultra-fast synthesis).”

Comment 8. On page 10, what is meant by “magnifying the faint changes of nanozymes”?

Response. We are grateful for this comment. One of the goals of our work is to examine the physiochemical changes of PBNZ after catalysis, which is beneficial for their mechanism study. However, such changes cannot be clearly detected and are frequently neglected when the catalytic duration is not long enough. For example, a previous work

claimed that the oxidation of PBNZ was not observed after treated with H₂O₂ in 60 minutes (*Nano Research*, 11, (2018), 4905-4913). However, the oxidation of PBNZ was confirmed by conducting the long-term catalysis in our work. The minor change of zeta potential, optical spectrum, valence state and catalytic activity after short-term catalysis might be hard to measure, while the strategy of long-term catalysis is believed to magnify these physicochemical changes and make them detectable. An explicit discussion on the advantages of the long-term catalysis were added in the Introduction section (**page 4-5, line 83-88**) and the repeated expression in the Results and discussion section were deleted in the revised manuscript (**page 9, line 165-168**).

Page 4-5, line 83-88: “In fact, the long-term catalysis is advantageous to judge the working life span of nanozymes and amplify the minor physicochemical changes that may happen during the short-term catalysis which are faint and undetectable. We believe that this strategy is also helpful to examine the catalytic sustainability of PBNZ, detect the weak variation during the catalytic process and reveal their potential mechanism.”

Page 9, line 165-168: “Despite the undetected oxidation of PBNZ (treated with 20 mM H₂O₂ for 60 min) claimed by Dong’s group²⁵, our results revealed the actual existence of this irreversible oxidation process during the catalysis of PBNZ.”

Comment 9. It is not entirely clear if leaching of Fe is beneficial or detrimental. It seems like both are claimed but maybe I misunderstood.

Response. Thank you very much for your careful review. As shown in the

supplementary Fig. 12, the Fe leaching happened when the PBNZ were dispersed in HAc-NaAc buffer and the oxidation effect of H₂O₂ will facilitate the leaching during the cyclic catalysis of PBNZ or their direct incubation with H₂O₂. Based on the ICP-MS and XPS results, the amorphous 81 nm and 43 nm PBNZ could be easily oxidized by H₂O₂ with a higher leaching concentration of Fe. While the crystalline 90 nm PBNZ could be hardly oxidized with a lower leaching of Fe. Therefore, the Fe leaching could to some extent represent the oxidation degree of PBNZ: a higher concentration of Fe leaching means a higher oxidation degree of PBNZ with stronger catalytic activity. However, the excessive Fe leaching may reduce the surficial catalytic sites of PBNZ, which is detrimental for their catalytic activity enhancement. As a result, the advantage and disadvantage of Fe leaching may simultaneously exist. A revised discussion on the influence of Fe leaching on the catalytic activity of PBNZ was shown in the Supplementary discussion for Supplementary Fig. 12 (paragraph 1) in SI and highlighted in red.

Supplementary Fig. 12 Fe leaching effect on the POD-like activity of PBNZ.

Concentration of Fe element in the leaching solution of **a** 81 nm, **b** 43 nm and **c** 90 nm PBNZ catalytic system. Error bars represent the standard deviation of ICP-MS measurement. **d** POD-like activity of leaching solution obtained from the incubation of PBNZ with H₂O₂. Error bars represent standard deviation from three independent measurements.

Supplementary discussion for Supplementary Fig. 12, paragraph 1: “The leaching solution of PBNZ catalytic system was obtained by ultrafiltration to remove the nanoparticles. The concentration of leaching Fe element was measured by ICP-MS. An obvious growth of Fe element content was observed when PBNZ were incubated with

0.12% H₂O₂ or in cyclic catalysis day 1, while the increment of Fe leaching in the first round of cyclic catalysis was relatively lower (Supplementary Fig. 12a-c). Therefore, it is inferred that the Fe leaching could reflect the oxidation of PBNZ from another perspective: a higher amount of Fe leaching represents a higher oxidation degree. Therefore, the higher degree of Fe leaching found in the system of lowly crystalline PBNZ than that in highly crystalline 90 nm PBNZ reflected the ease of oxidation for PBNZ with low crystallinity. Of note, excessive Fe leaching would cause the loss of active sites on the surface of PBNZ, which detrimentally reduced the increment of their catalytic activity. As a result, although a stronger oxidation effect occurred in 43 nm PBNZ because of their larger specific surface area, the POD-like activity increment of 43 nm PBNZ was not obviously larger than 81 nm PBNZ (Fig. 2b, c).”

Comment 10. If there is substantial leaching, how can be made sure that the catalytic process occurs on the PBNZ nanoparticle and is not catalyzed by Fe ions in solution as well?

Response. Thank you sincerely for your insightful question. We firstly estimated the POD and CAT-like catalytic ability of the leaching Fe ions obtained from the long-term catalysis through the following steps: PBNZ were incubated with H₂O₂ in HAc-NaAc buffer (for POD-like activity) or water (for CAT-like activity) for 24 h (the concentration of PBNZ and H₂O₂ were the same as described in the methods section of long-term catalysis). The leaching solutions were then obtained by ultrafiltration. Finally, the POD-like activity of the leaching solutions was measured by a direct

addition of 0.385 mg/mL ABTS and 0.12% H₂O₂. The CAT-like activity of the leaching solutions was measured by the direct addition of 3.6% H₂O₂.

The obtained leaching solution showed a faster absorbance change compared with the control group in the POD-like activity measurement (Supplementary Fig. 12d). Similarly, the higher oxygen concentration also reflected a certain CAT-like activity of the leaching solution (Supplementary Fig. 16b). Thus, it can be concluded that the PBNZ and leaching Fe ions would both contribute for the catalytic activity during the prolonged catalysis. Notably, the catalytic activity of leaching solution would be exaggerated by the current measurement method, as the ABTS was not added during the incubation and the residual H₂O₂ in the solution was not removed before the measurement. Moreover, apparently reduced Fe leaching was observed when the recycled PBNZ were redispersed into the buffer and experienced a new round of cyclic catalysis (Supplementary Fig. 12a-c). This means that the catalytic contribution of the leaching solution in prolonged catalysis will continuously decrease as the round of cyclic catalysis increases, ensuring that the catalytic process could mostly proceed on the PBNZ nanoparticles in the subsequent cyclic catalysis.

Nevertheless, the main idea for the design of long-term catalysis is to judge the catalytic activity variation of the recycled PBNZ which was commonly measured in five minutes. To evaluate the catalytic contribution of leaching Fe in short-term catalytic process, leaching solutions were obtained by ultrafiltration after the incubation of PBNZ with H₂O₂ for 5 minutes. As shown in Supplementary Fig. 12d and Supplementary Fig. 16b, the catalytic activity of the leaching solution obtained from 5

min incubation was very weak. Therefore, although the gradual Fe leaching during the long-term catalysis could compete with the nanoparticles and partially contribute for the catalytic activity, such homogeneous catalysis effect can be ignored during the short-term catalysis. In other words, it is ensured that the short-term catalysis happens on the PBNZ nanoparticles, which will not interfere their catalytic activity measurement and the following mechanism study. A revised discussion on the influence of Fe leaching on the catalytic activity of PBNZ was shown in the paragraph 2-3 of Supplementary discussion for Supplementary Fig. 12 and Supplementary discussion for Supplementary Fig. 16 in SI (highlighted in red).

Supplementary Fig. 12 Fe leaching effect on the POD-like activity of PBNZ.

Concentration of Fe element in the leaching solution of **a** 81 nm, **b** 43 nm and **c** 90 nm PBNZ catalytic system. Error bars represent the standard deviation of ICP-MS measurement. **d** POD-like activity of leaching solution obtained from the incubation of PBNZ with H₂O₂. Error bars represent standard deviation from three independent measurements.

Supplementary discussion for Supplementary Fig. 12, paragraph 2-3: “Since the Fe leaching exists in the long-term catalysis of PBNZ, it is vital to evaluate the catalytic contribution of the leaching Fe ions. The POD-like catalytic ability of the leaching Fe ions obtained from the long-term catalysis was estimated through the following steps:

4.6 $\mu\text{g}/\text{mL}$ PBNZ were incubated with 0.12% H_2O_2 in HAc-NaAc buffer for 24 h. The POD-like activity of the obtained leaching solutions was measured by a direct addition of ABTS and H_2O_2 . As seen in Supplementary Fig. 12d, the leaching solution showed a faster absorbance change compared with the control group. Thus, it can be concluded that both of the PBNZ and leaching Fe ions contributed for the catalytic activity during the prolonged catalysis. Notably, the catalytic activity of leaching solution would be exaggerated by the current measurement method, as the ABTS was not added during the incubation and the residual H_2O_2 in the solution was not removed before the measurement. Moreover, apparently reduced Fe leaching was observed when the recycled PBNZ were redispersed into the buffer and experienced a new round of cyclic catalysis (Supplementary Fig. 12a-c). This means that the catalytic contribution of the leaching solution in prolonged catalysis will continuously decrease as the round of cyclic catalysis increase, ensuring that the catalytic process could mostly proceed on the PBNZ nanoparticles even in long-term catalysis.

Nevertheless, the main idea for the design of long-term catalysis is to judge the catalytic activity variation of the recycled PBNZ which was commonly measured in only one minutes. To evaluate the catalytic contribution of leaching Fe in short-term catalytic process, leaching solutions were obtained by ultrafiltration after the incubation of PBNZ with H_2O_2 for 5 minutes. As shown in Supplementary Fig. 12d, the catalytic activity of the leaching solution obtained from 5 min incubation was very weak. Therefore, although the gradual Fe leaching during the long-term catalysis could compete with the nanoparticles and partially contribute for the catalytic activity, such

homogeneous catalysis effect can be ignored during the short-term catalysis. In other words, it is ensured that the short-term catalysis happens on the PBNZ nanoparticles, which will not interfere their catalytic activity measurement and the following mechanism study.”

Supplementary Fig. 16 Fe leaching effect on the CAT-like activity of PBNZ. **a** Concentration of Fe element in leaching solution. Error bars represent the standard deviation of ICP-MS measurement. **b** CAT-like activity of leaching solution obtained from the incubation of PBNZ with H₂O₂. Error bars represent standard deviation from three independent measurements.

Supplementary discussion for Supplementary Fig. 16: An obvious growth of Fe element content was observed when PBNZ were incubated with 3.6% H₂O₂ in pure water (Supplementary Fig. 16a). Similar to the situation in POD-like catalysis (Supplementary Fig. 12), the leaching Fe ions contributed for a certain amount of CAT-like activity in long-term catalysis (24 h) but would not interfere the CAT-like activity measurement of PBNZ in short term (5 min) for the following mechanism study.

Comment 11. The reported decrease in bandgap upon H₂O₂ oxidation is only 0.01 eV, which is within the error of the fitting.

Response. We are grateful for this comment. The bandgap was measured by the optical spectrum which reflects the bandgap variation in the bulk phase of recycled PBNZ. However, the oxidation of PBNZ during the catalysis only occurred on their surface. Therefore, such surficial bandgap value (E_g) variation would be underestimated by bulk phase bandgap measurement, leading to the seemingly minor decrease of E_g during the actual detection.

Although a very slight decrease of E_g was measured after incubating PBNZ with 0.12% H₂O₂ in HAc-NaAc buffer, which could be suspected as a fitting error (Supplementary Fig. 20a, d), a larger E_g reduction could be found after the incubation of PBNZ with 3.6% H₂O₂ in pure water (Supplementary Fig. 20c, f). Moreover, we further measured the E_g change of PBNZ after their incubation with 3.6% H₂O₂ in HAc-NaAc buffer, a larger decrease of E_g was also observed (Supplementary Fig. 20b, e). Taken together, the bandgap decrease of the oxidized PBNZ indeed existed as it became more obvious as the concentration of H₂O₂ increased and is not due to the fitting error.

Reported work has compared the E_g of PB, PY and PW with the relationship as followed: PB < PY << PW (*Nano Energy* 78 (2020) 105148). However, the actual existing N-coordinated Fe (II) (Fig.2 b, e) leads to the appearance of partial PW structures on the surface of PBNZ. Therefore, the minor decrease of E_g could be explained by the oxidation effect of H₂O₂ which improved the surficial N-coordinated Fe (III) content. The explanation for the bandgap decrease of the oxidized PBNZ was

revised and highlighted in red (Supplementary discussion for Supplementary Fig. 20) in the supplementary information.

Supplementary Fig. 20 Optical band gap of PBNZ before and after the oxidation of H_2O_2 . **a-c** 81 nm PBNZ. **d-f** 43 nm PBNZ.

Supplementary discussion for Supplementary Fig. 20: The E_g decrement after the surface oxidation of PBNZ by 0.12% H_2O_2 was quite weak (Supplementary Fig. 20a, d), which could be suspected as an error of fitting. However, a further decrease was observed when the dosage of H_2O_2 was raised to 3.6% in the acidic or neutral environment, confirming the regulating effect of oxidation on the band gap of PBNZ (Supplementary Fig. 20b, c, e and f). Reported work has compared the E_g value of PB, PY and PW with the relationship as followed: $PB < PY \ll PW^3$. However, the actual existing N-coordinated Fe (II) leads to the appearance of partial PW structures on the surface of PBNZ. Therefore, the minor decrease of E_g could be explained by the oxidation effect of H_2O_2 which transform the surficial structure of PW to PB. Besides,

the oxidation mainly happened on the surface of particles, while the optical method was used to measure the E_g in the bulk phase of PBNZ. Thus, such bulk phase measurement would underestimate the variation of the surficial E_g , resulting in the seemingly low decrement of bandgap during the actual detection.

Reviewer #2 (Remarks to the Author):

The manuscript entitled ‘Elucidating the catalytic mechanism of Prussian blue nanozymes with self-increasable activity from valence state and energy level perspectives’ reported the self-enhancing activity of Prussian blue nanozymes (PBNZ) acting as peroxidase (POD) and catalase (CAT) mimetics during prolonged catalysis. The unclear catalytic process of PBNZ was explored from the perspectives of valence state and energy level, supported by designed experimental evidence. The proposed dual-path electron transfer process contributed to a valuable understanding on the catalytic mechanism of nanozymes. The manuscript can be considered to be published after addressing the following concerns.

Response. Thank you sincerely for your positive comments.

Comment 1. I recommend emphasizing the catalytic mechanism of the dual-path electron transfer process throughout the entire manuscript. The current version lacks

focus. Additionally, it is advisable to include more results related to the catalytic mechanism, such as model construction and simulation of the catalytic process.

Response. Thank you very much for your professional suggestion. We have thoroughly revised the writing of our manuscript to highlight the dual-path electron transfer catalytic mechanism, which is the core discovery of this work. The concept and validation of the proposed mechanism were emphasized throughout the entire manuscript, while other data and relative discussions that are not crucial for the storyline were transferred to the Supplementary Information file. Moreover, we have conducted the model construction and DFT calculation of the catalytic process to further support our experimental conclusions. Detailed revisions were listed as below and highlighted in red in the revised manuscript:

(1) Brief conclusion of the dual-path electron transfer mechanism in the Abstract section (**Page 2, line 28-34**): “It is demonstrated that the irreversible oxidation of PBNZ is prominent for the promotion of catalysis and makes their catalytic activities self-increasable. The catalytic process of the pre-oxidized PBNZ could be initiated either through conduction band pathway or valence band pathway. Taken together, we firstly disclose that PBNZ follow a dual-path electron transfer mechanism during the POD and CAT-like catalysis with the superiority of long service life.”

(2) Introduction of the concept of valence band mediated electron transfer pathway (VBP) and conduction band mediated electron transfer pathway (CBP) in the Introduction section (**Page 4, line 67-76**): “Despite of the complexity, the overall electron flow direction of POD-like catalysis is from reductive substrate (e.g., 2,2'-

Azinobis (3-ethylbenzothiazoline -6-sulfonic Acid Ammonium Salt) (ABTS)) to H_2O_2 . However, two distinct electron transfer pathways could simultaneously exist: as a semiconductor³³, PB could undergo a valence band mediated pathway (VBP) where PB priorly donates an electron to H_2O_2 (process 1) and then accepts another electron from ABTS (process 2) (Fig. 1a); while the catalytic reaction could also occur through a conduction band mediated pathway (CBP) where PB or their pre-oxidized state (BG, PY) firstly receive an electron from ABTS (process 1), following with an electron transfer to H_2O_2 (process 2) (Fig. 1b)”

Fig. 1 Two possible electron transfer pathways during the POD-like catalysis of PBNZ. **a** Electron transfer pathway I (VBP). **b** Electron transfer pathway II (CBP).

(3) Validation of the dual-path electron transfer mechanism in the section of ‘Catalytic mechanism of PBNZ as POD and CAT mimetics’.

(i) Identification of the active intermediates (Fe-OH and Fe=O) during the catalysis of PBNZ (Page 13-14, line 248-270).

(ii) Estimation on the occurrence possibility of the VBP and CBP (Page 15-16, line 271-294).

(iii) Model construction and DFT calculation of the catalytic process (Page 16-18, line

295-339): “Density functional theory (DFT) calculations were conducted to understand the atomistic-level mechanism of the POD-like activity for PBNZ and the role of oxidation in enhancing their activity. According to the experimental results, Fe-OH and Fe=O are probably the key intermediates for the catalysis. Therefore, three kinds of PBNZ structures were comparatively calculated: pristine PBNZ (A1 of Fig. 5c), PBNZ modified with -OH (B1), and those with =O (C1 and D1). For a given PBNZ structure, the catalysis may start by reacting firstly with H₂O₂ (via the VBP) or ABTS (via the CBP). The calculated results suggested that the pristine PBNZ A1 had little POD-like activity. Instead, it could be easily oxidized by H₂O₂ to form a thermally stable intermediate (A3), which had a deep potential energy well (−3.25 eV) and was hard to be reduced back to A1 by ABTS (Fig. 5c). Further calculations suggested that the two-OH structure A3 could dehydrate to form a one-O structure (D1), with an energy change of 0.7 eV. With more OH groups on PBNZ, the dehydration became more energetically favorable. The dehydration of three-OH structure B3 to one-OH-and-one-O structure C2 was energetically favorable with an energy change of −0.11 eV. Therefore, the concentration of Fe=O would increase with the extent of oxidation. These results are consistent with the above experiments that unoxidized PBNZ had a lower POD-like activity and that Fe-OH and Fe=O groups were abundant in the PBNZ structures because of the irreversible oxidation effect.

In contrast, the oxidized PBNZ structure B1 had a shallow potential energy well and should have considerable POD-like activity. As shown in Figure 5c, one-OH structure B1, which represented the moderately oxidized PBNZ, could catalyze the

reaction between H_2O_2 and ABTS via the VBP. First, a H_2O_2 molecule dissociated on B1 to form B3 with an energy decrease of -0.76 eV, and then B3 was reduced back to B1 by two ABTS molecules, completing the catalytic cycle. Two-O structure C1, which represented the heavily oxidized PBNZ, could undergo the POD-like catalysis via the CPB. First, two-O structure C1 was reduced by two ABTS to form one-O structure C3 with an energy decrease of -0.25 eV, and then C3 was oxidized back to C1 by H_2O_2 (Figure 5d). One-O structure D1 could not undergo the POD-like catalysis via the CBP because of the high energy structure D3 (2.6 eV) involved in this pathway. For the same reason, three-OH structure B3 could catalyze the reaction via the CBP (B3-B4-B1-B2-B3) and one-O structure could catalyze the reaction via the VBP (C3-C4-C1-C2-C3). Notably, the Fe=O exists in both of the POD-like process mediated by PBNZ and horseradish peroxidase (HRP) with two sequential one-electron injections from the chromogenic substrates, which to some extent shows their similarity. However, the concrete variation process of Fe=O during their catalysis is quite different (PBNZ: $\text{Fe}=\text{O} \rightarrow \text{Fe}-\text{OH} \rightarrow \text{Fe}$; HRP: $\cdot\text{Fe}=\text{O}^+ \rightarrow \text{Fe}=\text{O} \rightarrow \text{Fe}^{42}$). Besides, since the CAT-like process could be regarded as a special POD-like process by replacing the ABTS with the reductive H_2O_2 under neutral pH, the above conclusion could be also applicable for the explanation of the CAT-like activity enhancement of the oxidized PBNZ. Taken together, the unoxidized PBNZ (e.g., A1) had little catalytic activity; while once the PBNZ were dispersed in the catalytic solution and irreversibly pre-oxidized by H_2O_2 , the moderately and heavily oxidized PBNZ exhibited the considerable POD and CAT-like activity via the VBP and CBP, respectively.”

Fig. 5 Dual-path electron transfer mechanism of PBNZ. **a** PMSO and PMSO₂ detection by HPLC. **b** Energy levels of catalytic substrates and conduction/valence band of PBNZ. Error bars represent standard deviation from three independent measurements. **c** Mechanism and energy profiles (energy in eV) mediated by VBP. **d** Mechanism and energy profiles (energy in eV) mediated by CBP.

(4) Brief conclusion of the dual-path electron transfer mechanism at the end of the manuscript (Page 19, line 350-354): “Due to the pre-oxidation effect of H₂O₂, the valence state of Fe on the surface of PBNZ increases with the formation of oxygenated groups, which is essential for the promotion of VBP and CBP during the catalysis. The

as-proposed dual-path electron transfer mechanism comprehensively elucidates the catalytic behavior of PBNZ with a long service life.”

Comment 2. In Fig. 3b, the TEM image suggests the presence of aggregated 43 nm PBNZ. How does this aggregation potentially affect their catalytic behavior?

Response. Thank you sincerely for your insightful question. Although aggregation was observed in the TEM images of these irregularly small PBNZ (Supplementary Fig. 2b), their DLS size was stably measured as 43 nm which indicated that the 43 nm PBNZ could be well dispersed in the solution (Supplementary Fig. 2a). Thus, the aggregation phenomenon only occurred during the drying process during sample preparation for TEM observation and will not affect their catalytic behavior in solution, which was also proved in our previous work (*ACS Applied Nano Materials* 4 (2021) 5176-5186). An explanation for this aggregation phenomenon was added and highlighted in red in the Supplementary discussion for the Supplementary Fig. 2.

Supplementary Fig. 2 Synthesis and characterization of 43 nm PBNZ. a DLS

measurement. **b** TEM and electron diffraction images. **c** UV-vis spectrum. **d** Element mapping images.

Supplementary discussion for Supplementary Fig. 2: As the x value increased to 20.0 mL, PBNZ with a hydrodynamic size of 43 ± 2.9 nm and zeta potential of -37.1 ± 0.6 mV were obtained (Supplementary Fig. 2a). The TEM and electronic diffraction images demonstrated their irregular morphology with low crystallinity (Supplementary Fig. 2b). **Notably, the aggregation of PBNZ observed in the TEM images was attributed to the drying process during the sample preparation for TEM observation (the particle size is thus hard to measure) and will not affect their catalytic behavior in solution¹.** The maximum UV-vis absorption peak of 43 nm PBNZ was also located near the wavelength of 700 nm (Supplementary Fig. 2c). The element mapping analysis revealed the existence of K, Fe and N elements in the particles (Supplementary Fig. 2d).

Comment 3. The authors noted an increased leakage of Fe in Fig. 7 and Fig. 9 following the incubation of PBNZ with H₂O₂. Please discuss the contribution of Fe leaching to POD- and CAT-like catalysis?

Response. We are grateful for the reviewer's comment. We firstly estimated the POD and CAT-like catalytic ability of the leaching Fe ions obtained from the long-term catalysis through the following steps: PBNZ were incubated with H₂O₂ in HAc-NaAc buffer (for POD-like activity) or water (for CAT-like activity) for 24 h (the concentration of PBNZ and H₂O₂ were the same as described in the methods section of long-term catalysis). The leaching solutions were then obtained by ultrafiltration.

Finally, the POD-like activity of the leaching solutions was measured by a direct addition of 0.385 mg/mL ABTS and 0.12% H₂O₂. The CAT-like activity of the leaching solutions was measured by the direct addition of 3.6% H₂O₂.

The obtained leaching solution showed a faster absorbance change compared with the control group in the POD-like activity measurement (Supplementary Fig. 12d). Similarly, the higher oxygen concentration also reflected a certain CAT-like activity of the leaching solution (Supplementary Fig. 16b). Thus, it can be concluded that the PBNZ and leaching Fe ions would both contribute for the catalytic activity during the prolonged catalysis. Notably, the catalytic activity of leaching solution would be exaggerated by the current measurement method, as the ABTS was not added during the incubation and the residual H₂O₂ in the solution was not removed before the measurement. Moreover, apparently reduced Fe leaching was observed when the recycled PBNZ were redispersed into the buffer and experienced a new round of cyclic catalysis (Supplementary Fig. 12a-c). This means that the catalytic contribution of the leaching solution in prolonged catalysis will continuously decrease as the round of cyclic catalysis increases, ensuring that the catalytic process could mostly proceed on the PBNZ nanoparticle in the subsequent cyclic catalysis.

Nevertheless, the main idea for the design of long-term catalysis is to judge the catalytic activity variation of the recycled PBNZ which was commonly measured in five minutes. To evaluate the catalytic contribution of leaching Fe in short-term catalytic process, leaching solutions were obtained by ultrafiltration after the incubation of PBNZ with H₂O₂ for 5 minutes. As shown in Supplementary Fig. 12d and

Supplementary Fig. 16b, the catalytic activity of the leaching solution obtained from 5 min incubation was very weak. Therefore, although the gradual Fe leaching during the long-term catalysis could compete with the nanoparticles and partially contribute for the catalytic activity, such homogeneous catalysis effect can be ignored during the short-term catalysis. In other words, it is ensured that the short-term catalysis happens on the PBNZ nanoparticles, which will not interfere their catalytic activity measurement and the following mechanism study. A revised discussion on the influence of Fe leaching on the catalytic activity of PBNZ was shown in the paragraph 2-3 of Supplementary discussion for Supplementary Fig. 12 and Supplementary discussion for Supplementary Fig. 16 in SI (highlighted in red).

Supplementary Fig. 12 Fe leaching effect on the POD-like activity of PBNZ.

Concentration of Fe element in the leaching solution of **a** 81 nm, **b** 43 nm and **c** 90 nm PBNZ catalytic system. Error bars represent the standard deviation of ICP-MS measurement. **d** POD-like activity of leaching solution obtained from the incubation of PBNZ with H₂O₂. Error bars represent standard deviation from three independent measurements.

Supplementary discussion for Supplementary Fig. 12, paragraph 2-3: “Since the Fe leaching exists in the long-term catalysis of PBNZ, it is vital to evaluate the catalytic contribution of the leaching Fe ions. The POD-like catalytic ability of the leaching Fe ions obtained from the long-term catalysis was estimated through the following steps:

4.6 $\mu\text{g}/\text{mL}$ PBNZ were incubated with 0.12% H_2O_2 in HAc-NaAc buffer for 24 h. The POD-like activity of the obtained leaching solutions was measured by a direct addition of ABTS and H_2O_2 . As seen in Supplementary Fig. 12d, the leaching solution showed a faster absorbance change compared with the control group. Thus, it can be concluded that both of the PBNZ and leaching Fe ions contributed for the catalytic activity during the prolonged catalysis. Notably, the catalytic activity of leaching solution would be exaggerated by the current measurement method, as the ABTS was not added during the incubation and the residual H_2O_2 in the solution was not removed before the measurement. Moreover, apparently reduced Fe leaching was observed when the recycled PBNZ were redispersed into the buffer and experienced a new round of cyclic catalysis (Supplementary Fig. 12a-c). This means that the catalytic contribution of the leaching solution in prolonged catalysis will continuously decrease as the round of cyclic catalysis increases, ensuring that the catalytic process could mostly proceed on the PBNZ nanoparticles in the subsequent cyclic catalysis.

Nevertheless, the main idea for the design of long-term catalysis is to judge the catalytic activity variation of the recycled PBNZ which was commonly measured in only one minutes. To evaluate the catalytic contribution of leaching Fe in short-term catalytic process, leaching solutions were obtained by ultrafiltration after the incubation of PBNZ with H_2O_2 for 5 minutes. As shown in Supplementary Fig. 12d, the catalytic activity of the leaching solution obtained from 5 min incubation was very weak. Therefore, although the gradual Fe leaching during the long-term catalysis could compete with the nanoparticles and partially contribute for the catalytic activity, such

homogeneous catalysis effect can be ignored during the short-term catalysis. In other words, it is ensured that the short-term catalysis happens on the PBNZ nanoparticles, which will not interfere their catalytic activity measurement and the following mechanism study.”

Supplementary Fig. 16 Fe leaching effect on the CAT-like activity of PBNZ. **a** Concentration of Fe element in leaching solution. Error bars represent the standard deviation of ICP-MS measurement. **b** CAT-like activity of leaching solution obtained from the incubation of PBNZ with H₂O₂. Error bars represent standard deviation from three independent measurements.

Supplementary discussion for Supplementary Fig. 16: An obvious growth of Fe element content was observed when PBNZ were incubated with 3.6% H₂O₂ in pure water (Supplementary Fig. 16a). Similar to the situation in POD-like catalysis (Supplementary Fig. 12), the leaching Fe ions contributed for a certain amount of CAT-like activity in long-term catalysis (24 h) but would not interfere the CAT-like activity measurement of PBNZ in short term (5 min) for the following mechanism study.

Comment 4. When discussing the Fe (IV) process in PBNZ catalysis, it is recommended to describe the mechanism of horseradish peroxidase, which can underscore the similarities between PBNZ and natural enzymes.

Response. Thank you sincerely for this helpful suggestion. The Fe (IV)=O exists in the both of the POD-like process mediated by PBNZ and horseradish peroxidase (HRP), showing the similarity to some extent. However, the concrete variation process of Fe (IV)=O during their catalysis is different. A brief comparison between the catalytic mechanism of horseradish peroxidase and PBNZ was added when discussing the Fe (IV)=O process in PBNZ catalysis in the revised manuscript and highlighted in red.

Page 17, line 327-332: “Notably, the Fe=O exists in the both of the POD-like process mediated by PBNZ and horseradish peroxidase (HRP) with two sequential one-electron injections from the chromogenic substrates, which to some extent shows their similarity. However, the concrete variation process of Fe=O during their catalysis is quite different (PBNZ: $\text{Fe}=\text{O} \rightarrow \text{Fe}-\text{OH} \rightarrow \text{Fe}$; HRP: $\cdot\text{Fe}=\text{O}^+ \rightarrow \text{Fe}=\text{O} \rightarrow \text{Fe}^{42}$).”

Comment 5. The energy band gap narrowing of PBNZ after oxidation was relatively weak. A more detailed explanation of this phenomenon should be provided.

Response. We are grateful for this comment. The bandgap change was measured by optical spectrum which reflects the bandgap variation in the bulk phase of recycled PBNZ. However, the oxidation of PBNZ during the catalysis only occurred on their surface. Therefore, such surficial bandgap value (E_g) variation would be underestimated by the bulk phase bandgap measurement, leading to the seemingly minor decrease in

bandgap during the actual detection.

Although a very slight decrease of E_g was measured after incubating PBNZ with 0.12% H_2O_2 in HAc-NaAc buffer, which could be suspected as a fitting error (Supplementary Fig. 20a, d), a larger E_g reduction could be found after the incubation of PBNZ with 3.6% H_2O_2 in pure water (Supplementary Fig. 20c, f). Moreover, we further measured the E_g change of PBNZ after their incubation with 3.6% H_2O_2 in HAc-NaAc buffer, a larger decrease of E_g was also observed (Supplementary Fig. 20b, e). Taken together, the bandgap decrease of the oxidized PBNZ indeed existed as it became more obvious as the concentration of H_2O_2 increased and is not due to the fitting error.

Reported work has compared the E_g of PB, PY and PW with the relationship as followed: $PB < PY \ll PW$ (*Nano Energy* 78 (2020) 105148). However, the actual existing N-coordinated Fe (II) (Fig.2 b, e) leads to the appearance of partial PW structures on the surface of PBNZ. Therefore, the minor decrease of E_g could be explained by the oxidation effect of H_2O_2 which improved the surficial N-coordinated Fe (III) content. The explanation for the bandgap decrease of the oxidized PBNZ was revised and highlighted in red (Supplementary discussion for Supplementary Fig. 20) in the supplementary information.

Supplementary Fig. 20 Optical band gap of PBNZ before and after the oxidation of H_2O_2 . **a-c** 81 nm PBNZ. **d-f** 43 nm PBNZ.

Supplementary discussion for Supplementary Fig. 20: The E_g decrement after the surface oxidation of PBNZ by 0.12% H_2O_2 was quite weak (Supplementary Fig. 20a, d), which could be suspected as an error of fitting. However, a further decrease was observed when the dosage of H_2O_2 was raised to 3.6% in the acidic or neutral environment, confirming the regulating effect of oxidation on the band gap of PBNZ (Supplementary Fig. 20b, c, e and f). Reported work has compared the E_g value of PB, PY and PW with the relationship as followed: $\text{PB} < \text{PY} \ll \text{PW}^3$. However, the actual existing N-coordinated Fe (II) leads to the appearance of partial PW structures on the surface of PBNZ. Therefore, the minor decrease of E_g could be explained by the oxidation effect of H_2O_2 which transform the surficial structure of PW to PB. Besides, the oxidation mainly happened on the surface of particles, while the optical method was used to measure the E_g in the bulk phase of PBNZ. Thus, such bulk phase measurement

would underestimate the variation of the surficial E_g , resulting in the seemingly low decrement of bandgap during the actual detection.

Comment 6. In the Fe (IV) detection using 90 PBNZ (Fig. 8a), the transformation efficiency of PMSO/PMSO₂ was quite low and further decreased during the detection in neutral pH (Fig. 10a). The reasons underlying this observation need to be explained.

Response. We are grateful for this professional suggestion. As discussed in **page 10--11, line 193-202** and Supplementary Fig. 11, the 90 nm PBNZ were harder to be oxidized compared with the 81 nm and 43 nm PBNZ. In other words, high valent Fe was more difficult to be generated through the oxidation of crystalline 90 nm PBNZ by H₂O₂, causing the lower transformation efficiency of PMSO/PMSO₂ in the detective system of 90 nm PBNZ (Supplementary Fig. 17).

The observed decreasing amount of PMSO₂ in neutral pH was partly attributed to the reduced oxidability of H₂O₂ as the pH increase from the acidic to the neutral, which directly decreased the production of Fe=O. Another reason is the competitive CAT-like reaction under neutral pH, which would largely consume the Fe=O to produce O₂. As demonstrated in Supplementary Fig. 18, only a slight decrease of the CAT-like activity of 81 nm and 43 nm PBNZ was observed after adding the excessive amount of PMSO. In other words, the generated Fe=O would be consumed by the oxidation of H₂O₂ to form O₂ under neutral pH, leading to the decreased generation of PMSO₂. As a result, an increased portion of PMSO was oxidized by Fe-OH instead.

Taken together, the evaluation on the transformation efficiency of PMSO/PMSO₂

could primarily confirm the coexistence of Fe=O and Fe-OH during the catalysis of PBNZ. We have revised the discussion on the PMSO oxidation experiment in the revised manuscript and supplementary information.

Page 14, line 250-267: “Active intermediate identification was firstly conducted using methyl phenyl sulfoxide (PMSO) as the detective probe. Briefly, PMSO could be specifically oxidized by Fe-OH and Fe=O respectively, the latter oxidation process leads to the formation of PMSO₂ with a light adsorption at the wavelength of 215 nm. Thus, the generation of Fe=O during the catalysis could be quantitatively measured by calculating the transformation efficiency (η) of PMSO₂ from PMSO⁴⁰. As shown in Fig. 5a, both of the 81 nm and 43 nm PBNZ detective systems exhibited high η value at pH 3.6, revealing the large formation of Fe=O. A lower η value ($57.0\% \pm 2.7\%$) was observed in the 90 nm PBNZ detective system (Supplementary Fig. 17), again demonstrating the difficulty of oxidation in 90 nm PBNZ. The η value showed a rapid decrease in pure water (pH 7.0), the observed decreasing amount of PMSO₂ was partly attributed to the reduced oxidability of H₂O₂ under neutral pH, which directly decreased the production of Fe=O²⁴. Another reason is the competitive CAT-like reaction under neutral pH, which would largely consume the Fe=O to produce O₂ (Supplementary Fig. 18). From another perspective, the incomplete transformation of PMSO₂ also reflected the existence of Fe-OH during the catalysis of PBNZ, which was further confirmed by electron spin resonance spectrometer (EPR) (Supplementary Fig. 19a).”

Supplementary Fig. 17 PMSO and PMSO₂ detection by HPLC in 90 nm PBNZ system.

Error bars represent standard deviation from three independent measurements.

Supplementary discussion for Supplementary Fig. 17: As discussed in the previous section, the 90 nm PBNZ were harder to be oxidized compared with the 81 nm and 43 nm PBNZ. In other words, high valent Fe was more difficult to be generated through the oxidation of crystalline 90 nm PBNZ by H₂O₂, causing the lower transformation efficiency of PMSO/PMSO₂ in the detective system of 90 nm PBNZ.

Supplementary Fig. 18 Competitive consumption of the Fe=O by CAT-like catalysis and oxidation of PMSO. Error bars represent standard deviation from three independent measurements.

Supplementary discussion for Supplementary Fig. 18: Only a slight decrease of the

CAT-like activity of 81 nm and 43 nm PBNZ was observed after adding the excessive amount of PMSO. In other words, the generated Fe=O would be consumed by the reduction of H₂O₂ to form O₂ under neutral pH, leading to the decreased generation of PMSO₂. As a result, an increased portion of PMSO was oxidized by Fe-OH rather than Fe=O.

Comment 7. The main manuscript includes an abundance of figures. The dual-path electron catalytic mechanism and the supported figures needed to be highlighted. Other figures are suggested to move to the supplementary information.

Response. Thank you very much for this suggestion. Figures that are not crucial for the storyline in the main manuscript were transferred to the supplementary information and renumbered as Supplementary Fig. 1, Fig.2, Fig. 6, Fig. 7, Fig. 10, Fig. 12, Fig. 13, Fig. 14, Fig. 16, Fig. 17, Fig. 18 and Fig. 20 with supplementary discussions. The number of figures in the main manuscript was reduced to five with magnified size and reduced subfigures which highlighted the dual path electron transfer catalytic mechanism (Fig. 1-5).

Comment 8. The whole manuscript should undergo proofreading, and any grammar mistakes should be corrected.

Response. Thank you very much for pointing out this problem. To make the manuscript more readable, the grammar and expression mistakes throughout the manuscript were carefully corrected and highlighted in blue in the revised manuscript.

Reviewer #3 (Remarks to the Author):

The authors present an interesting and original study about the catalytic activity of Prussian blue nanozymes from a mechanistic approach. The manuscript is interesting in the field but there are some parts difficult to follow mainly due to inappropriate use of the language.

Response. Thank you very much for your encouraging comments. The detailed corrections on language expression are shown in the following responses.

Comment 1. Lines 344-345: Given that Fe (II) dominated the surface, their surface state was more like PW.

Response. Thank you for your comments. The main reason for the high surficial Fe(II)/Fe(III) ratio of PBNZ is the largely existing C-coordinated Fe(II), thus the expression ‘their surface state was more like PW’ is not appropriate. However, the XPS results revealed a certain proportion of N-coordinated Fe(II), indicating the surficial existence of PW structure (Fe(II)[Fe(II)(CN)₆]²⁻) which increases the bandgap of the as-prepared PBNZ. This inappropriate expression was corrected and highlighted in red in the Supplementary discussion for Supplementary Fig. 20.

Supplementary Fig. 20 Optical band gap of PBNZ before and after the oxidation of H_2O_2 . **a-c** 81 nm PBNZ. **d-f** 43 nm PBNZ.

Supplementary discussion for Supplementary Fig. 20: The E_g decrement after the surface oxidation of PBNZ by 0.12% H_2O_2 was quite weak (Supplementary Fig. 20a, d), which could be suspected as an error of fitting. However, a further decrease was observed when the dosage of H_2O_2 was raised to 3.6% in the acidic or neutral environment, confirming the regulating effect of oxidation on the band gap of PBNZ (Supplementary Fig. 20b, c, e and f). Reported work has compared the E_g value of PB, PY and PW with the relationship as followed: $PB < PY \ll PW^3$. However, the actual existing N-coordinated Fe (II) leads to the appearance of partial PW structures on the surface of PBNZ. Therefore, the minor decrease of E_g could be explained by the oxidation effect of H_2O_2 which transform the surficial structure of PW to PB. Besides, the oxidation mainly happened on the surface of particles, while the optical method was used to measure the E_g in the bulk phase of PBNZ. Thus, such bulk phase measurement

would underestimate the variation of the surficial E_g , resulting in the seemingly low decrement of bandgap during the actual detection

Comment 2. Line 381. The Apparent physicochemical properties change.... Refer to the changes described in the previous paragraph?

Response. Thank you for your careful review. The mentioned ‘Apparent physicochemical properties change’ refers to the changes that happened in the PBNZ recycled from prolonged CAT-like catalysis. These changes are similar to the previous changes as described in the PBNZ recycled from long-term POD-like catalysis. We apologize for this inaccurate expression. The modified expression was listed as below and highlighted in red in revised manuscript.

Page 12, line 222-223: “Similar to the prolonged POD-like catalysis, apparent physicochemical changes were also observed in the PBNZ recycled from the long-term CAT-like reaction.”

Comment 3. Lines 386-387: Color visibly changed, how? Decreased characteristic absorption at 700 nm, means that the maximum is displaced or that there is a decrease in the absorption intensity?

Response. We are grateful for this comment. We are sorry for the unclear explanation for the color change of the PBNZ recycled from prolonged CAT-like catalysis. Compared with the adsorption spectrum of the original PBNZ, the adsorption intensity around 700 nm in the spectrum of the recycled PBNZ decreased while the adsorption

intensity among 300-500 nm increased (Fig. 4c). The above adsorption change is consistent with the spectrum variation during the transformation of Prussian blue to Berlin green or Prussian yellow in the previous reports (*Advanced Functional Materials* 14 (2004) 224-232, *Thin Solid Films* 789 (2024) 140192). Therefore, the slight green color of the recycled PBNZ solution (Supplementary Fig. 14) was attributed to the oxidation of PBNZ during the catalysis. The revised discussion on the color and spectrum change of PBNZ was highlighted in red in the revised manuscript and supplementary information.

Fig. 4 Characterization of the PBNZ recycled from CAT-like long-term catalysis.

a Hydrodynamic size, **b** zeta potential, **c** UV-vis spectra, **d**, **e** TEM images, **f**, **g** fitting

XPS Fe 2p spectrum and **h** relative CAT-like activity of the recycled PBNZ. Error bars represent standard deviation from three independent measurements.

Supplementary Fig. 14 Photos of PBNZ before and after the prolonged CAT-like catalysis.

Page 12, line 225-228: “The color of the recycled PBNZ solution become slightly green (especially the 43 nm PBNZ) (Supplementary Fig. 14) accompanied with the decreased adsorption intensity around 700 nm and increased adsorption intensity among 300-500 nm (Fig. 4c).”

Comment 4. Conclusion’s section is not clear. Authors conclude in lines 364, 421 and 444 (probably the real conclusion part), corresponding to the last part of each section. Sentence of lines 444 to 446 is not clear.

Response. Thank you very much for this comment. We are sorry for the redundant conclusive expressions. The repeated conclusions in our original manuscript were deleted. The aim of the sentence in line 444 to 446 in the original manuscript is to emphasize the advantage of the long-term catalysis experiment, which could help discover the faint physiochemical changes and study the catalytic mechanism of

nanozymes. We have clarified our conclusion in the final part of the revised manuscript (highlighted in red).

Page 19, line 346-354: “In summary, the catalytic mechanism of PBNZ is clarified through the experimental and theoretical analyses on the PBNZ recycled from long-term catalysis. Our results reveal the unique advantage of PBNZ with self-increasable POD and CAT-like activities compared with natural enzymes and other depletable iron-based nanozymes. Due to the pre-oxidation effect of H₂O₂, the valence state of Fe on the surface of PBNZ increases with the formation of oxygenated groups, which is essential for the promotion of VBP and CBP during the catalysis. The as-proposed dual-path electron transfer mechanism comprehensively elucidates the catalytic behavior of PBNZ with a long service life.”

Comment 5. Line 453, what is the meaning of The above findings are instructive for life span judgement?

Response. We are grateful for this question. In this work, the conducted long-term catalysis experiment revealed the self-increasable catalytic activity of PBNZ, indicating their high catalytic sustainability as POD and CAT mimetics. The original aim of the sentence “The above findings are instructive for life span judgement” is to emphasize the advantage of long-term catalysis experiment, which is beneficial for the study of the catalytic sustainability and working life span of nanozymes. However, since we have already discussed the superiority of long-term catalysis experiment in the Introduction section (**Page 4-5, line 83-88**), we have deleted this elusive expression in

the conclusion section for the conciseness of the manuscript.

Minor remarks:

1. All over the text nanozymes tested are of 81 and 43 nm but in the method section, line 478, describes the synthesis of 90 and 43 nm.

Response. We are grateful for this careful review. The 81 and 43 nm PBNZ are the main analytic objects in this work, therefore we deleted the synthetic description of 90 nm PBNZ in the method section. However, the highly crystalline 90 nm PBNZ are also important for the catalytic mechanism study. Thus, a brief description on the synthesis process was given before discussing the crystallinity effect on the oxidation of PBNZ by H₂O₂.

Page 20, line 378-380: “Briefly, the *x* value was set as 1.0 mL and 20.0 mL to obtain 81 nm PBNZ and 43 nm PBNZ with low crystallinity, respectively. The reaction time was set as 25 min (81 nm PBNZ) and 2h (43 nm PBNZ).”

Page 10-11, line 193-197: “To study the influence of crystallinity on the oxidation of PBNZ, the highly crystalline PBNZ (denoted as 90 nm PBNZ) were synthesized under the *x* value of 1.0 mL and the reaction time of 2h, which exhibited similar particle size, surficial valence state and POD-like activity but different crystallinity compared with the 81 nm PBNZ (Supplementary Fig. 11a-f).”

2. Figure 3b) should show the size as in figure 2b.

Response. We are grateful for this suggestion. Unfortunately, although the 43 nm PBNZ showed high dispersity in solution, these nanoparticles tended to aggregate in

the drying process during the sample preparation for TEM observation. Therefore, it is hard to calculate the accurate size of 43 nm PBNZ from their TEM image. The relevant explanation was added in the revised supplementary information and highlighted in red.

Supplementary Fig. 2 Synthesis and characterization of 43 nm PBNZ. **a** DLS measurement. **b** TEM and electron diffraction images. **c** UV-vis spectrum. **d** Element mapping images.

Supplementary discussion for Supplementary Fig. 2: As the x value increased to 20.0 mL, PBNZ with a hydrodynamic size of 43 ± 2.9 nm and zeta potential of -37.1 ± 0.6 mV were obtained (Supplementary Fig. 2a). The TEM and electronic diffraction images demonstrated their irregular morphology with low crystallinity (Supplementary Fig. 2b). **Notably, the aggregation of PBNZ observed in the TEM images was attributed to the drying process during the sample preparation for TEM observation (the particle size is thus hard to measure) and will not affect their catalytic behavior in solution¹.** The maximum UV-vis absorption peak of 43 nm PBNZ was also located near the wavelength of 700 nm (Supplementary Fig. 2c). The element mapping analysis revealed

the existence of K, Fe and N elements in the particles (Supplementary Fig. 2d).

3. Line 139, result should be omitted.

Response. Thank you for your careful review. We have deleted the word 'result' in the revised manuscript.

4. Figure 4. Colors of the lines for 43 and 81 nm are confusing.

Response. We are grateful for this comment. We have changed the colors of the lines for 43 and 81 nm PBNZ as shown in the revised Fig. 2g-i.

Fig. 2 Surface composition and catalytic activity of prepared PBNZ. a XPS survey,

b Fe 2p spectrum and **c** EDS spectrum of 81 nm PBNZ. **d** XPS survey, **e** Fe 2p spectrum and **f** EDS spectrum of 43 nm PBNZ. POD-like activity measurement using **g** ABTS and **h** TMB as substrates. **i** CAT-like activity measurement. Error bars represent standard deviation from three independent measurements.

Reviewers' Comments:

Reviewer #1:

Remarks to the Author:

The authors answered my questions well and have largely improved the manuscript according to my suggestions. It is now more readable and concise.

However, I would still recommend giving the text to a native English speaker to read. While the text has improved, many expressions still do not sound correct.

I still think that some subfigures could be moved to the SI or summarized, such as the many XPS and EDS data, but this is only a minor issue.

The authors have now added DFT calculations to the manuscript, which, in general, is a good idea and may support the conclusions. However, there is not enough information to evaluate if the DFT calculations are done well and if they can be trusted.

Some questions and concerns from my side are:

1) How exactly was the slab constructed? How many layers does it have? And, importantly, how was the surface cut?

The Fe-CN bond (cyanide coordinating from its C-side) is a very strong chemical bond. If this bond is cut when generating the surface, the surface will be unstable and not represent a viable model.

2) The authors should show the used surface models in the SI. The construction and optimisation process of the chosen surface should be explained. I also recommend uploading the optimised geometries of all structures (bulk, surfaces and surfaces with adsorbates) on an openly accessible database, such as the NOMAD or the ioChemBD repositories.

3) The reaction energies given are DFT energies and not Gibbs free energies if I understand it correctly. One should, however, give the free energy differences, including entropic and zero-point vibrational energy contributions.

4) Are other reaction pathways, adsorption sites, etc. explored?

5) Importantly, what happens to the spin on the Fe sites and the adsorbates? It may be possible that the active Fe centre, and even neighbouring centres, may transition between high-spin, low-spin, and intermediate-spin states. Was the spin included in the calculation of the reaction pathway?

Computing magnetic moments in Prussian blue is very complex and needs very careful attention.

Reviewer #2:

Remarks to the Author:

The revised manuscript have addressed my concern, and I have no additional comments

Reviewer #3:

Remarks to the Author:

The authors have answered the questions posed by the reviewer appropriately and the manuscript can be considered for publication.

Dear reviewers,

We sincerely thank you for your constructive comments. We have revised and improved our manuscript according to your suggestions. Our detailed, point-by-point responses are listed in the following pages. A copy tracking all the changes made in our revision has also been submitted. We hope that our responses could be acceptable.

Sincerely yours,

Yu Zhang

Southeast University, Nanjing 211102, P. R. China

E-mail address: zhangyu@seu.edu.cn

Point-by-point responses to the reviewers' comments:

Reviewer #1 (Remarks to the Author):

The authors answered my questions well and have largely improved the manuscript according to my suggestions. It is now more readable and concise.

Response. We are grateful for your careful review and positive comment.

However, I would still recommend giving the text to a native English speaker to read.

While the text has improved, many expressions still do not sound correct.

Response. We appreciate the reviewer's suggestion. We have passed the manuscript to

a native English speaker to improve our writing. The inappropriate expressions and grammar mistakes have been corrected and highlighted in blue in the revised manuscript.

I still think that some subfigures could be moved to the SI or summarized, such as the many XPS and EDS data, but this is only a minor issue.

Response. Thank you very much for your suggestion. Although a certain amount of XPS and EDS data was presented in the main manuscript, we think that these data are important for revealing the neglected physiochemical properties of PBNZ and surficial changes during the catalysis. As these data are quite crucial for the storyline, we are afraid that it is better to maintain them in the main article.

The authors have now added DFT calculations to the manuscript, which, in general, is a good idea and may support the conclusions. However, there is not enough information to evaluate if the DFT calculations are done well and if they can be trusted. Some questions and concerns from my side are:

1) How exactly was the slab constructed? How many layers does it have? And, importantly, how was the surface cut?

The Fe-CN bond (cyanide coordinating from its C-side) is a very strong chemical bond. If this bond is cut when generating the surface, the surface will be unstable and not represent a viable model.

Response. Thank you very much for kindly reminding us to include the important

computational details in the Method section. The slab model was constructed as follows. First, the bulk structure of $\text{KFe(III)[Fe(II)(CN)}_6]$, which contained four K-atoms, eight Fe-atoms, and twenty four CN-groups [$\text{K}_4\text{Fe}_8(\text{CN})_{24}$], was built and geometrically relaxed using PBE+U method (Supplementary Fig. 21). On the basis of the relaxed bulk structure, the (1×1) PBNZ slab was cut along the (001) direction (Supplementary Fig. 21b). The PBNZ (001) slab also contained four K-atoms, eight Fe-atoms, and twenty four CN-groups [$\text{K}_4\text{Fe}_8(\text{CN})_{24}$]. The four inner Fe atoms were six-coordinated; the four surface Fe atoms were five-coordinated. The PBNZ (001) slab contained eight atomic layers, and atoms in the bottom four layers were fixed during the subsequent calculations. The magnetic moments of C- and N-coordinated Fe on the surface were $0.052 \mu_B$ and $4.381 \mu_B$, respectively, which were close to the corresponding magnetic moments of Fe in the lattice.

We also agree with you that the CN^- group had a strong coordinating interaction with the surface Fe atoms. However, under the circumstances of H_2O_2 , the surface CN^- groups should be replaced by the more reactive OH groups. Indeed, our calculations suggested that the adsorption energy for OH on surface Fe(II) site was -3.29 eV in the water condition, whereas that for CN^- was only -1.14 eV . Such calculated result was consistent with the XPS experiment (Supplementary Fig. 15), which showed that oxygen species were introduced into the material sample. Therefore, our slab models with OH or O groups which terminated the surface Fe atoms were reasonable.

According to the above comment, the following statements on computational details have been added to the revised manuscript and Supplementary Information

(highlighted in red):

Page 24, line 465-467: “More details of the calculations can be found in the supplementary computational details (Supplementary Fig. 21 and Supplementary Fig. 22).”

Supplementary Fig. 21 Construction of PBNZ (001) slab model. **a** Bulk structure with electron spin configurations for the N- and C-coordinated Fe atoms. **b** Side view of PBNZ (001) slab.

Supplementary discussion for supplementary Fig. 21: The bulk structure of $\text{KFe(III)[Fe(II)(CN)}_6]$, which contained four K-atoms, eight Fe-atoms, and twenty four CN-groups [$\text{K}_4\text{Fe}_8(\text{CN})_{24}$], was built and geometrically relaxed using the PBE+U method (Supplementary Fig. 21a). The model was similar with previous theoretical studies and real experiment structure^{4,5}. The N-coordinated Fe (Fe^{3+}) located at the corners of face-centered cubic (fcc) lattice of the crystal group. The C-coordinated Fe (Fe^{2+}) located at the middle of edge. The difference in the valence state of Fe was compensated by K^+ , locating in half of the tetrahedral holes in the lattice. The

calculation results showed that the cell parameters of $\text{K}_4\text{Fe}_8(\text{CN})_{24}$ were: $a = b = c = 10.261 \text{ \AA}$ and $\alpha = \beta = \gamma = 90^\circ$. These values were close to the experimental cell parameters of PBNZ⁴. The magnetic moments of N- and C-coordinated Fe were $4.467 \mu_{\text{B}}$ and $0.107 \mu_{\text{B}}$, attributed to the high-spin Fe^{3+} and low-spin Fe^{2+} , respectively. This was consistent with the experimental magnetic data of PBNZ⁶.

On the basis of the relaxed bulk structure, the (1×1) PBNZ slab was cut along the (001) direction (Supplementary Fig. 21b). The PBNZ (001) slab also contained four K-atoms, eight Fe-atoms, and twenty four CN-groups [$\text{K}_4\text{Fe}_8(\text{CN})_{24}$]. The four inner Fe atoms were six-coordinated; the four surface Fe atoms were five-coordinated. The PBNZ (001) slab contained eight atomic layers, and atoms in the bottom four layers were fixed during the subsequent calculations. The magnetic moments of C- and N-coordinated Fe on the surface were $0.052 \mu_{\text{B}}$ and $4.381 \mu_{\text{B}}$, respectively, which were close to the corresponding magnetic moments of Fe in the lattice. Additionally, H_2O_2 can easily oxidize PB to form $\cdot\text{OH}$. The $\cdot\text{OH}$ had an adsorption energy of -3.29 eV on surface Fe^{2+} site, whereas CN^- had only an adsorption energy of -1.14 eV . Thus, $\cdot\text{OH}$ also bonded to Fe site on the surface by replacing CN^- . Therefore, the selected PBNZ (001) was a reasonable catalytic reaction model.

2) The authors should show the used surface models in the SI. The construction and optimisation process of the chosen surface should be explained. I also recommend uploading the optimised geometries of all structures (bulk, surfaces and surfaces with adsorbates) on an openly accessible database, such as the NOMAD or the ioChemBD

repositories.

Response. Thank you for your kind suggestion. Please see the above response for the computational details. Besides, we have uploaded all the structure files in a zip file (named as 'Source data') for the ease of review and we will upload them on an openly accessible database if no further revision is required.

3) The reaction energies given are DFT energies and not Gibbs free energies if I understand it correctly. One should, however, give the free energy differences, including entropic and zero-point vibrational energy contributions.

Response. Thank you very much for your professional comment. We agree with you that Gibbs free energy is the exact physical quantity for reaction thermodynamics. However, for reactions occurring on solid surfaces, the changes in entropy and zero-point energy are usually minor, except for those in the adsorption and desorption processes. Because the key reactions we studied were on the surface, the total energies were almost as equivalent as Gibbs free energies. Therefore, total energies were used throughout our work. The similar strategy has been widely used for the computations of surface reactions (e.g., ACS Catal. 2016, 6, 8370–8379; J. Am. Chem.Soc. 2012, 134, 1560–1570; J. Am. Chem. Soc. 2016, 138, 2629–2637.).

4) Are other reaction pathways, adsorption sites, etc. explored?

Response. Thank you very much for your valuable question. We have taken a comprehensive consideration on the possible reaction pathways and adsorption sites.

Experimentally, the PBNZ surfaces were covered by OH and O groups that came from the dissociation of H₂O₂. The numbers and proportions of these oxygen groups greatly affected the POD-like activity of PBNZ. We have considered PBNZ slabs with different degrees of oxidation, which include the pristine unoxidized PBNZ slab (Supplementary Fig. 21) and four oxidized slabs (Supplementary Fig. 22). Given that the reducing ability of Fe(II) is stronger than Fe(III), OH or O groups were preferentially added on the Fe(II) sites when constructing the models. We assumed that OH and O adsorbed on the Fe(II) site and formed the Fe(III) and Fe(IV) sites, respectively. These five slabs were used as initial structural models for investigating the catalytic pathways. The mechanisms and energy profiles of the five structures were already shown in the manuscript (Fig. 5c, d) except for E1. In fact, E1 also exhibited POD-like activity via VBP pathway (Supplementary Fig. 23). Considering the complexity of the reaction system, we only showed some of the typical reaction pathways in our original manuscript.

According to this comment, we have added the structural models of the four oxidized slabs and E1 initiated catalytic pathway in the revised manuscript and supplementary information (highlighted in red):

Page 24, line 465-467: “More details of the calculations can be found in the supplementary computational details (Supplementary Fig. 21 and Supplementary Fig. 22).”

Supplementary Fig. 22 Construction of the PBNZ (001) slab with O or OH group.

Page 17, line 317-318: “Similarly, the two-OH structure E1 as shown in Supplementary Fig. 23 could catalyze the reaction via the VBP, the three-OH structure B3 could catalyze the reaction via the CBP (B3-B4-B1-B2-B3), and the one-O structure could catalyze the reaction via the VBP (C3-C4-C1-C2-C3).”

Supplementary Fig. 23 Energy profiles (energy in eV) of VBP initiated by E1.

5) Importantly, what happens to the spin on the Fe sites and the adsorbates? It may be possible that the active Fe centre, and even neighbouring centres, may transition between high-spin, low-spin, and intermediate-spin states. Was the spin included in the calculation of the reaction pathway?

Computing magnetic moments in Prussian blue is very complex and needs very careful attention.

Response. Thank you for your careful review and the important question on the electronic configuration of the material. We agree with you that the electronic configurations of PBNZ and its oxides should be always carefully checked, ensuring that the calculated electronic configuration is the ground state and the reactions are physically meaningful. In our calculations, the spin polarized density functional was applied to considered the spins. Briefly, the Fe(II) had a low spin state and the Fe(III) had a high spin state at the ground state, which were shown in Supplementary Fig. 21. The calculation results showed that the magnetic moments of N- and C-coordinated Fe in the bulk were $4.467 \mu_B$ and $0.107 \mu_B$, respectively. Therefore, they were regarded as the high-spin Fe(III) and low-spin Fe(II), respectively. The magnetic moments of C- and N-coordinated Fe on the surface were $0.052 \mu_B$ and $4.381 \mu_B$, respectively, which were close to the corresponding magnetic moments of Fe in the bulk. For Fe atoms that were attached with OH or O groups, the calculated magnetic moments were in between these two values. The detailed magnetic moments for the reaction species were given in the Additional Fig. 1 and Additional Table 1 for reviewer.

Additional Fig. 1 for reviewer. The structures in dual-path electron transfer mechanism of PBNZ. The yellow and green numbers in the figure represent Fe sites and adsorbates numbers, respectively.

Additional Table 1 for reviewer. Magnetic moment of the Fe sites and adsorbates in the dual-path electron transfer mechanism of PBNZ.

Structures	$\mu_B(\text{Fe1})$	$\mu_B(\text{Fe2})$	$\mu_B(\text{Fe3})$	$\mu_B(\text{Fe4})$	$\mu_B(\text{H}_2\text{O}_2)$	$\mu_B(\text{OH1})$	$\mu_B(\text{OH2})$	$\mu_B(\text{OH3})$	$\mu_B(\text{O1})$	$\mu_B(\text{O2})$
A1/D3	0.052	0.052	4.381	4.365						
A2/D4	4.308	0.024	3.822	3.824	0.058					
A3/C2	1.050	2.059	3.812	3.818		0.090	0.125			
A4/B1/D2	2.065	0.023	3.811	3.819		0.122				
B2	2.871	2.997	3.817	3.818	0.043	0.249				
B3	2.068	2.074	3.805	4.391		0.131	0.130	0.265		
B4	2.059	1.050	3.812	3.818		0.125	0.090			
C1	1.505	1.377	3.822	4.364					0.569	0.699
C3/D1	1.394	0.027	3.821	3.821					0.671	
C4	3.031	3.391	3.814	3.815	0.044				0.349	

According to this comment, we have added the following sentence to the computational methods in the revised manuscript and highlighted in red:

Page 24, line 468-469: “The spin polarized density functional theory calculations were applied to investigate reaction process.”

Reviewer #2 (Remarks to the Author):

The revised manuscript have addressed my concern, and I have no additional comments

Response. Thank you very much for your careful review and positive comment.

Reviewer #2 (Remarks on code availability):

I am not an expert on code aspect.

Reviewer #3 (Remarks to the Author):

The authors have answered the questions posed by the reviewer appropriately and the manuscript can be considered for publication.

Response. We appreciate for your careful review and positive comment.

Reviewers' Comments:

Reviewer #1:

Remarks to the Author:

The authors largely improved the manuscript, in particular, the English writing. They also answered all my questions regarding the computational part and added the required information to the manuscript and SI. I can now recommend the manuscript for publication.